# Iontronic pressure sensor with high sensitivity over ultra-broad linear range enabled by laser-induced gradient micro-pyramids

Ruoxi Yang[1,2], Ankan Dutta [2], Bowen Li [2], Naveen Tiwari[2], Wanqing Zhang [2], Zhenyuan Niu[2], Yuyan Gao[2], Daniel Erdely[2], Xin Xin[2], Tiejun Li[1,3] ✉ & Huanyu Cheng [2] ✉

Despite the extensive developments of flexible capacitive pressure sensors, it is still elusive to simultaneously achieve excellent linearity over a broad pressure range, high sensitivity, and ultrahigh pressure resolution under large pressure preloads. Here, we present a programmable fabrication method for microstructures to integrate an ultrathin ionic layer. The resulting optimized sensor exhibits a sensitivity of 33.7 kPa$^{-1}$ over a linear range of 1700 kPa, a detection limit of 0.36 Pa, and a pressure resolution of 0.00725% under the pressure of 2000 kPa. Taken together with rapid response/recovery and excellent repeatability, the sensor is applied to subtle pulse detection, interactive robotic hand, and ultrahigh-resolution smart weight scale/chair. The proposed fabrication approaches and design toolkit from this work can also be leveraged to easily tune the pressure sensor performance for varying target applications and open up opportunities to create other iontronic sensors.

The rapid developments of flexible pressure sensors have spurred new application opportunities for robotics[1], prosthetic[2], human-machine interface[3,4], and health monitoring[5]. Compared with the piezoelectric and piezoresistive pressure sensors, the capacitive pressure sensor is simple in structure, highly stable with low drift, and has low power consumption[6,7]. The practical applications of capacitive pressure sensors hinge on their sensing performance especially high sensitivity and linearity over a broad range for easy signal processing[8]. Among varying microstructures such as micro-dome[9] and hierarchical fabrics[10], the pyramidal structures[11,12] exhibit excellent compressibility and progressive deformation with increasing pressure to result in high sensitivity in a large linear range. The photolithography-based approaches have been mainly used to investigate the effects of various geometric parameters (e.g., sidewall angle, density, and frustum top/base length[13,14]). However, the high sensitivity drastically decreases

as the pressure further increases (to completely deform the pyramidal microstructures). There are opportunities to exploit non-uniformly distributed (e.g., gradient) microstructures, but the complex and expensive processes present challenges to exploring and optimizing the parameter space. As a result, microstructures have been templated from natural/existing objects (e.g., leaf[15,16], abrasive paper[17], petal[18], and fabric[19]), but the random microstructures result in uncontrollable deformation and narrow linear range. Therefore, it is important to design microstructure patterns[9] and/or varying permittivities at different microstructure locations[20] for increased linear range, though often at an expense of decreased sensitivity.

Besides modulating the overlapping area ($A$) and the distance ($d$) between two electrodes in the capacitive pressure sensor, it is of high interest to explore high dielectric ($\varepsilon$) materials for improved dielectric properties and increased sensitivity. Although much effort is

[1]School of Mechanical Engineering, Hebei University of Technology, 300401 Tianjin, China. [2]Department of Engineering Science and Mechanics, The Pennsylvania State University, University Park, PA 16802, USA. [3]School of Mechanical Engineering, Hebei University of Science & Technology, 050018 Shijiazhuang, China. ✉e-mail: 001036@hebust.edu.cn; huanyu.cheng@psu.edu

devoted to mixing high dielectric materials with nanomaterials (e.g., $CaCu_3Ti_4O_{12}$[21], MXenes[22], carbon nanotubes[23], and Ag nanowires[24]), the sensitivity of the capacitive pressure sensors is far from their piezoresistive counterparts. Atomically thin ionic liquids (ILs) layers (~1 nm) between positive and negative charges in the electron double layer (EDL)[25] at the electrode/dielectric interface can significantly increase the piezo-capacitive effect[26]. The EDL effect can be affected by the concentration of the ILs[17,27] and the measurement frequency[28]. Mixing ILs with high-polarity polymers (e.g., polyvinylidene fluoride (PVDF)-hexafluoropropylene copolymer (HFP)[17,29,30], thermoplastic polyurethane (TPU)[31]/polyurethane (PU)[32], and polyvinyl alcohol (PVA)[33,34]) to form a dielectric layer helps enhance the sensitivity of the microstructured pressure sensors[2,35,36]. Using electrode materials with ions intercalation-based pseudocapacitance such as $Ti_3C_2T_x$[33,37] in iontronic sensors can also enhance the capacitance and then the sensitivity. However, such high sensitivity is often limited to a small linear sensing range, with the sensitivity declining gradually as the pressure loading further increases.

High linearity over a broad sensing range usually comes at a cost of reduced sensitivity (e.g., 9280 $kPa^{-1}$ within 20 kPa[32] vs. 0.065 $kPa^{-1}$ within 1700 kPa[9]). The development of capacitive pressure sensors with high sensitivity in a wide linear range remains elusive due to the trade-off between sensitivity and linear range[29,30,34]. Efforts to address this challenge lead to the exploration of various structure designs in iontronic sensors. Representative examples include laminating multi-layer microstructured ionic layer (sensitivity of 9.17 $kPa^{-1}$ over 2000 kPa)[30] and adding microstructures on the surface of microdomes (sensitivity of 49.1 $kPa^{-1}$ within 485 kPa)[34]. Despite the rapid developments, fabrication of the critical microstructures still relies on complex fabrication processes such as photolithography or transferring of random microstructures. Therefore, there is an urgent demand for an effective and efficient fabrication method to prepare the programmable microstructure designed by quantitative numerical analysis.

This work explores the use of a $CO_2$ laser with a Gaussian beam profile to fabricate programmable structures such as gradient pyramidal microstructures (GPM) for iontronic sensors, reducing the cost and process complexity compared with photolithography. The profile and height of each pyramid in the gradient array can be individually adjusted and optimized to provide even deformation as the pressure increases. Combining the programmable microstructures with an ultrathin IL layer results in an iontronic sensor to break the trade-off between sensitivity and linear sensing range, exhibiting a high sensitivity (33.7 $kPa^{-1}$) and excellent linearity (0.99) over an ultra-broad linear sensing range up to 1700 kPa. The synergy between the programmed microstructure array and the enhanced electric field from IL also results in a low detection limit and high-pressure resolution under high pressure preload of 2000 kPa. As a result, the sensor is uniquely positioned to detect subtle pulses from the fingertip and tiny pressure variations under an ultrahigh pressure load with a smart weight scale, as well as to integrate with a control system for intelligent human-machine interactions. Taken together with the large-scale production capability, the facile design and fabrication framework from this work can also be applied to design and demonstrate smart prosthetic interfaces, intelligent human-machine interfaces, and next-generation health-monitoring devices.

## Results
### Fabrication and working mechanism
The flexible iontronic pressure sensor is designed to sandwich the iontronic dielectric layer between a bottom structured electrode and a top polyethylene terephthalate (PET)/indium tin oxide (ITO) electrode (Fig. 1a). The response of the capacitive iontronic pressure sensor results from the varied contact area between the structured pyramidal electrode and ion layer under compression. The good performance of the iontronic pressure sensor relies on a low-cost gradient pyramidal structure of laser-ablated polymethyl methacrylate (PMMA) that can

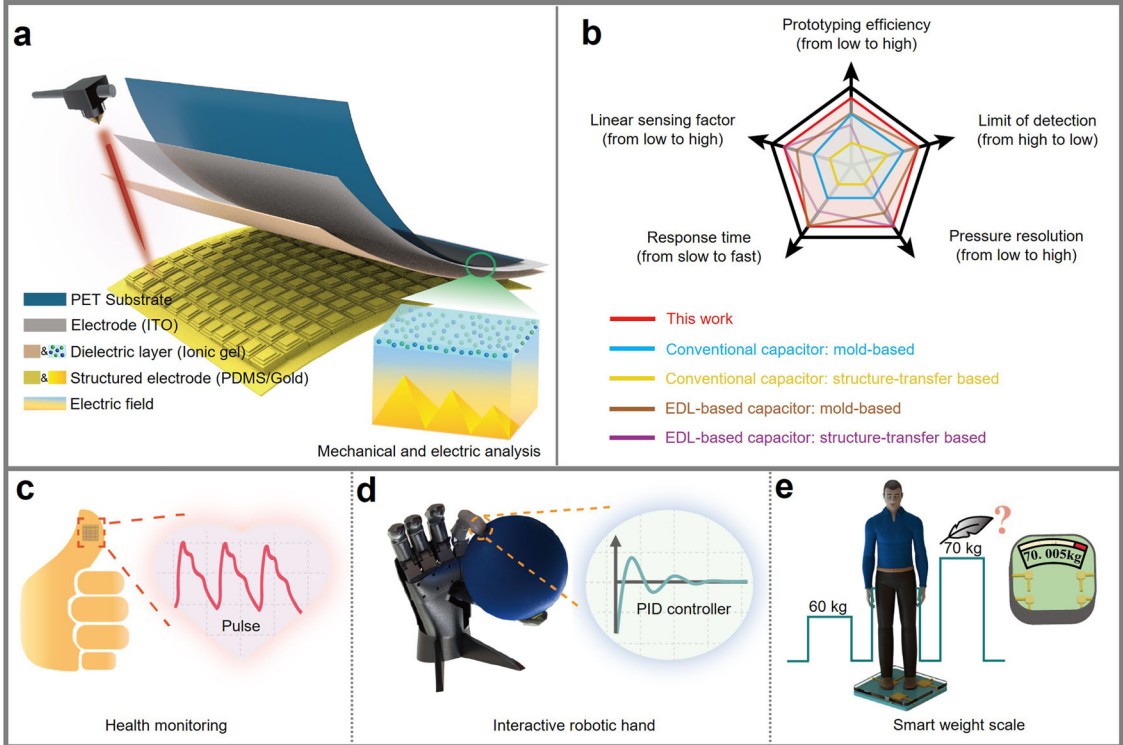

**Fig. 1 | Overview of the iontronic pressure sensor.** Schematic showing the **a** design, **b** performance, and application of the flexible iontronic pressure sensor in **c** health monitoring, **d** interactive robotic hand, and **e** smart weight scale. Note: the dielectric layer (pink) based on the iontronic liquid (IL) is sandwiched between the bottom gradient microstructured pyramidal electrode (yellow) and the top polyethylene terephthalate (PET)/indium tin oxide (ITO) electrode (gray). Linear sensing factor is defined as the product of the sensitivity and the corresponding linear sensing range.

be created by a multi-step process (Supplementary Table 1). The structured electrode detached from the PMMA mold can be simply assembled with the iontronic gel coated on the PET/ITO to yield the iontronic pressure sensor (Supplementary Fig. 1). The detailed fabrication process can be found in the "Methods" section. The gradient pyramidal electrode combined with the ultrathin ion gel provides the sensor with an ultrahigh sensitivity over an ultra-broad linear sensing range. Meanwhile, the fine features on the pyramidal microstructure contribute to the ultralow detection limit and ultrahigh pressure resolution under a large preload. Taken together with the fast response/recovery time, the flexible iontronic pressure sensor from this work outperforms both the microstructured conventional and EDL-based capacitive pressure sensors in the previous literature reports (Fig. 1b and Supplementary Table 2). The flexible iontronic pressure sensor with quality performance can be applied to a broad range of applications, including health monitoring, human-robot interaction/collaboration, and smart weight scale with ultrahigh pressure resolution (Fig. 1c–e).

After the external voltage is applied, electrons on the (top and bottom) electrodes attract the ions with opposite polarity in the iontronic layer to aggregate at the electrode/IL interfaces to form EDLs[38]. Numerous electron-ion pairs at the interface are separated by nanometer distance and behave as micro/nanoscale capacitors with ultrahigh capacitance. The capacitance at the interface is significantly increased as the microstructured electrode is deformed under pressure (Fig. 2a). The pyramids with different heights in the gradient microstructures as revealed by the scanning electron microscopy (SEM) (Fig. 2b–d) and optical images (Fig. 2e) can be sequentially deformed as the loading pressure increases, leading to a broad sensing range. Meanwhile, the gradient design also minimizes the initial contact area (before pressure) at the structured electrode/IL interface to result in a larger normalized contact area and capacitance change upon pressure loading. Therefore, the resulting sensor exhibits a much higher response than that with uniform pyramidal microstructures (UPM) or other microstructures (Fig. 2f), due to the reduced initial capacitance from 823 (square frustum) to 43 (UPM) and then 12 pF (gradient pyramidal microstructures with large height differences (GPML)). Increasing the concentration of the ions in the IL layer further enhances the EDL effect for increased pressure sensitivity, which can be combined with the gradient pyramidal microstructures with properly designed profiles for the iontronic pressure sensor to achieve high sensitivity over the ultra-broad range (Fig. 2g).

### Gradient pyramidal microstructures from laser-ablated molds
The one-step thermal decomposition and volatilization process of PMMA transforms it from solid to volatiles, which results from the C-O bond cleavage reaction, $[CH_2C(CH_3)(CO_2CH_3)]n{\rightarrow}CO + CO_2 + H_2O + H_2$. With a rectangular-gaussian[39] laser power intensity distribution (Supplementary Fig. 2a) that is sufficient to ablate PMMA, the profile of the laser-ablated holes appears to be Gaussian[40]. With different programmed laser patterns (Supplementary Table 3a–c), representative microstructures that are commonly used in pressure sensors can be easily fabricated, including conical frustums[41] (Supplementary Fig. 2b), cones[29] (Supplementary Fig. 2c), and square frustums[42] (Supplementary Fig. 2d). Compared with the circular laser patterns, the square laser patterns can create square frustum microstructures with high consistency.

As progressive deformation of the microstructures upon increasing pressure load can ensure high sensitivity over a wide sensing range, it is highly desirable to explore gradient microstructures. Careful design of the gradient can also give high linearity in the entire sensing range. However, fabrication is traditionally challenging even with complex photolithographic processes. Fortunately, the laser ablation of PMMA with the gaussian distributed laser beam provides a simple yet viable solution to overcome this challenge. Manipulating

the laser power represents the most intuitive way to control the structure height for obtaining gradient microstructures. Indeed, the continuously increased maximum power of the laser creates taller and slimmer microstructures (Supplementary Fig. 3a), but their high aspect ratio makes them susceptible to bulking upon pressure loading (Supplementary Fig. 3b). As a result, the nonlinear deformation from buckling[29,43,44] compromises the sensor performance in terms of linearity.

To address this issue, the multi-layered gradient pyramidal microstructures have been created by using multiple laser ablations, in which the successive laser ablation exhibits lower energy but wider distribution compared with the previous ablation (Supplementary Fig. 3c). The realization of pyramidal microstructures can be simply achieved by fixing the focus of the laser beam and reducing power from inside to outside of each pattern unit. The multi-layered microstructures with the same cross-sectional area are susceptible to bulking upon pressure loading (Supplementary Fig. 3d), so it is highly desirable to create those with a gradually reduced cross-sectional area. To achieve this, decreased laser power is accompanied by simultaneously increased square laser pattern in each successive layer for fabricating the tri-layered PMMA template and corresponding polydimethylsiloxane (PDMS) pyramidal microstructure (Supplementary Table 3d–f and Supplementary Fig. 4a–c). The pyramid-like microstructure with more accurate features is possible to achieve with an increased number of laser ablation layers. By modulating the sizes of the laser pattern while maintaining the ratio among three layers, different sizes of pyramidal microstructures can be formed (Supplementary Fig. 4d–f). Different heights in the resulting pyramidal microstructure can also be achieved by including an additional layer created from varying powers before laser ablating the tri-layered structures in the PMMA template (Supplementary Fig. 5a–c, see details in Supplementary Note 1 and Table 4). The microstructures in different layers can exhibit hierarchically similar features (Supplementary Fig. 5d).

### Sensing performance of different laser-induced structures
The deformation of the different microstructures upon pressure loading is first investigated by two-dimension (2D) finite element analysis (FEA) (Fig. 3a and Supplementary Movie 1). The pressure-induced capacitance change in the EDL-based capacitive sensor mainly comes from the changes in the contact area ($\Delta A/A_0$) at the electrode/dielectric interface. The change in the relative contact area or capacitance from the uniform microstructures with the same heights under increasing pressure indicates the highest sensitivity in the sensor with pyramidal structures due to the smallest tip area. Different from the UPM, the GPM is progressively compressed from the tall to the short ones as the pressure increases, resulting in the gradually increased contact area at the electrode/dielectric interface. The initially smaller contact area of the GPM contributes to larger changes and higher sensitivity (defined as the relative capacitance change $\Delta C/C_0$ over the applied pressure $P$) compared with the UPM under the same applied pressure. Additionally, the GPM with a larger height difference (e.g., GPML with 150 μm difference in $A_{300}B_{150}$) shows slightly higher sensitivity than that with a small height difference (e.g., GPMS with 50 μm difference in $A_{300}B_{250}C_{200}$) (Fig. 3b). The design insights and predictions from the FEA are validated by the experimental results with microstructures created from the laser-ablated PMMA molds. Compared with cone- and frustum-like microstructures, the pyramidal microstructure with a smaller tip area provides the sensor with the largest changes in the relative contact area and then capacitance upon pressure loading (Fig. 3c). Moreover, the GPML exhibits higher sensitivity than that of UPM and GPMS (Fig. 3d and Supplementary Note 2). As a result, the high sensitivity ($S = (\Delta C/C_0)/P$) of 2.49 kPa$^{-1}$ can be obtained in the GPML over an ultrawide range of 3000 kPa ($R^2 = 0.97$).

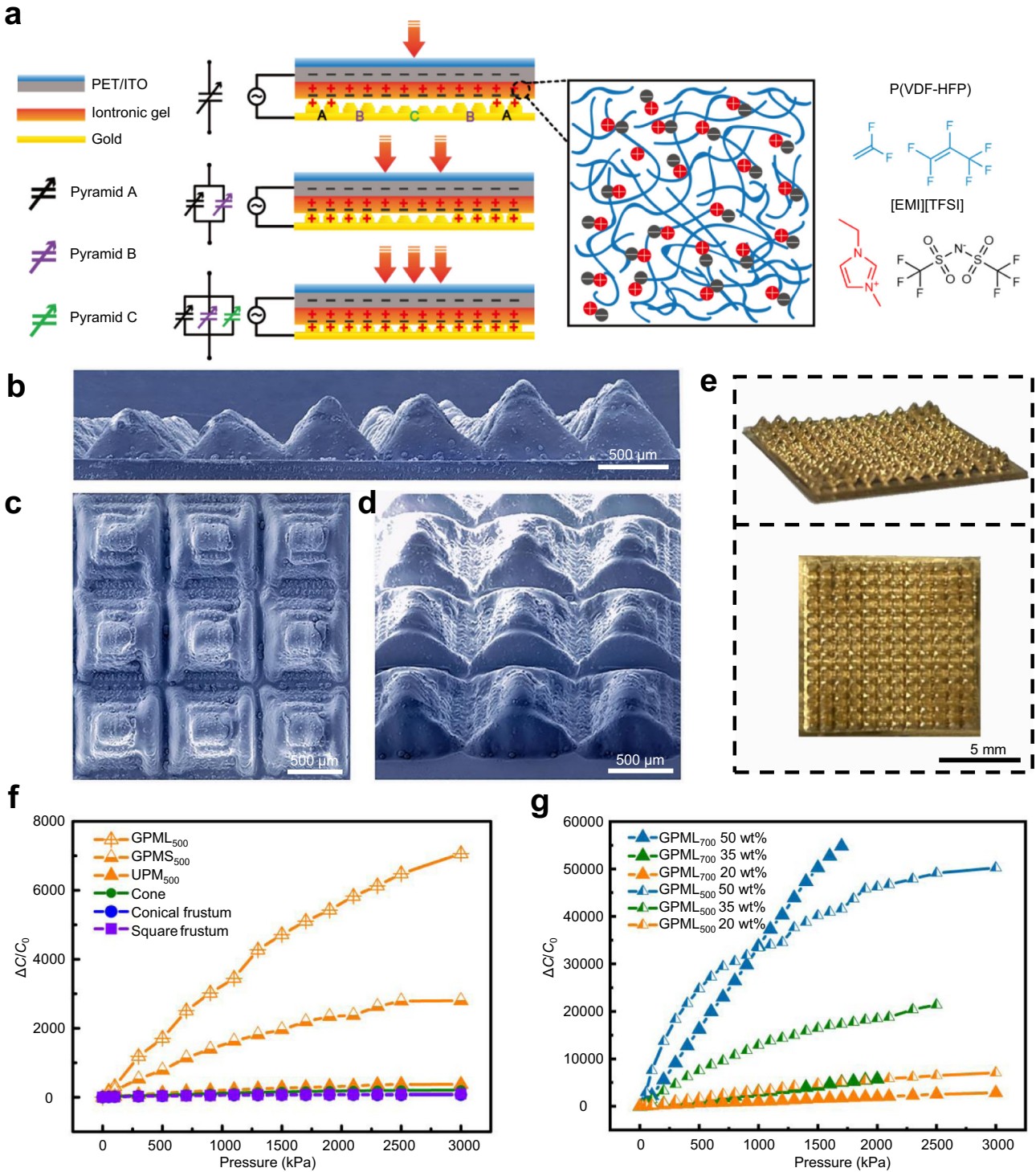

**Fig. 2 | Design and characterization of the iontronic pressure sensor based on gradient pyramidal microstructures. a** Schematic showing the working mechanism of the iontronic pressure sensor. Scanning electron microscopy (SEM) images of the gradient electrode in **b** side, **c** top, and **d** perspective views. **e** Optical images of the gradient electrode in tilt (top) and top (bottom) views. Comparison of the pressure sensing performance between the sensors with **f** different microstructures (20 wt% IL) and **g** different IL weight percentages. (UPM uniform pyramidal microstructures, GPM gradient pyramidal microstructures, GPML or GPMS gradient pyramidal microstructures with large or small height differences, the subscript represents the base length of laser-ablated patterns).

## Sensing performance modulated by the enhanced EDL effect

Modulation of the sensing performance can be easily achieved by increasing the concentration (20 wt%, 35 wt%, and 50 wt%) of the IL in the dielectric layer for an enhanced EDL effect. Increasing the IL concentration from 20 to 50 wt% results in an increased areal capacitance from 36.5 to 984 pF cm$^{-2}$ at 100 Hz (Fig. 4a). The capacitive measurements at lower frequencies (e.g., <1 kHz) are less stable than those at higher frequencies. Additionally, the rotation speed of the IL molecule at high frequencies becomes slower to give smaller capacitance and capacitance changes upon pressure loading. Therefore, the optimized measurement frequency of 1 kHz is chosen in the following studies unless specified otherwise (Fig. 4b).

The enhanced EDL effect can be revealed by the electric field (Fig. 4c) between the top electrode and microstructured bottom

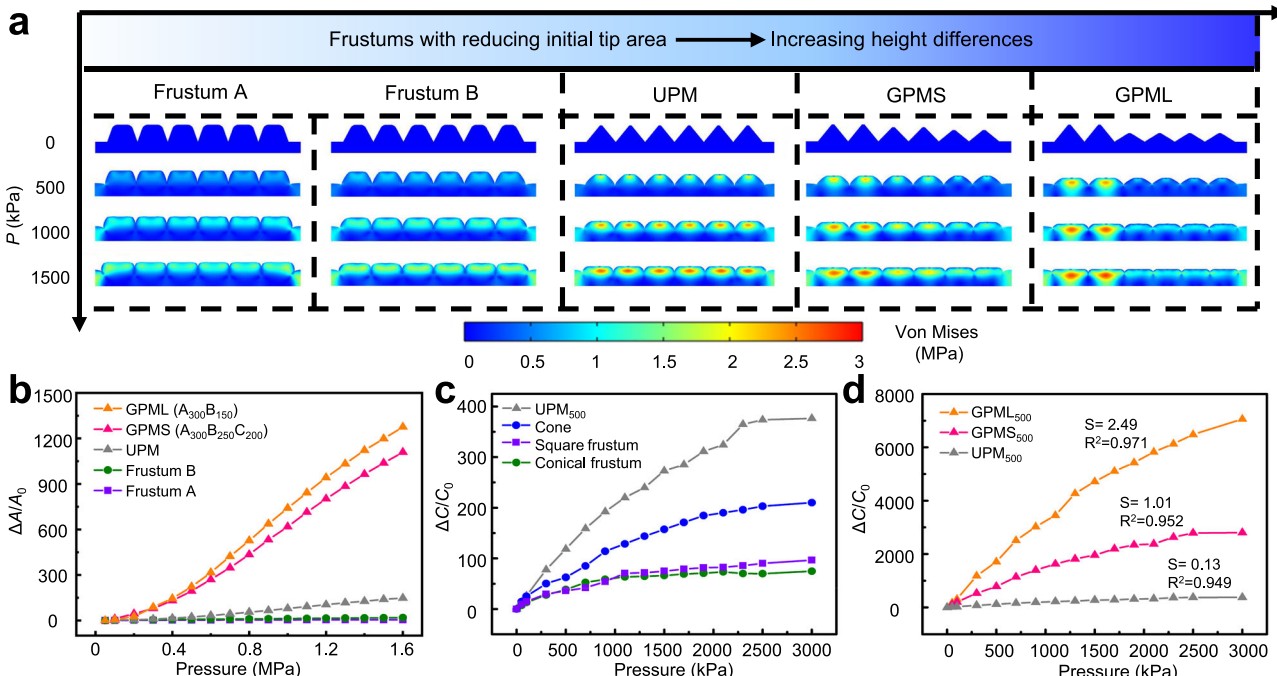

**Fig. 3 | Design of the iontronic pressure sensor with optimized microstructures.** Results from finite element analysis reveal the **a** stress distribution and **b** normalized relative contact area change ($\Delta A/A_0$) at the electrode/IL interface under pressure loading on the sensor with varying (uniform and gradient) microstructures. Experimental validation of the normalized relative capacitance change from the pressure sensor based on **c** uniform and **d** gradient microstructures (IL of 20 wt%).

electrode as the molarity of IL increases from 59 to 105 and then to 149 mol/m³ (Supplementary Table 5). Solving the Poisson, Nernst–Planck, and Butler–Volmer equations using COMSOL predicts a nonuniform potential distribution in the EDL domain (Supplementary Fig. 6a), resulting in a nonlinear electric field distribution (Fig. 4d). The nonlinearity of the electric field is also enhanced by the increased IL concentrations. The capacitance of the sensor increases because of the changes in the contact area from the mechanical deformation of pyramids under compression. However, as the pressure increases, the deformation of the pyramid gradually decreases to degrade the capacitance change rate and decrease the sensitivity. This issue can be effectively remedied by a stronger electric field at the later compression stage (with higher pressure loading) for increased capacitance, resulting in good linearity over the whole sensing range (Supplementary Fig. 6b). With the increased IL concentrations, the electric field is further enhanced to form the EDLs at the electrode/IL interface even if the two are not directly in contact[17]. However, further enhanced IL concentrations do not give an enhanced electric field upon higher pressure loading, resulting in increased sensitivity but decreased linear sensing range of the sensor.

The insights obtained from the simulation on the influence of the IL concentration are leveraged to optimize the performance of the iontronic pressure sensor in the experiment. Increasing the IL concentration in GPML$_{500}$ from 20 to 35 and then to 50 wt% leads to an increase of sensitivity from 2.49 to 13.1 and then to 43.6 kPa$^{-1}$. However, the linear sensing range is reduced from 3000 to 1200 and then to 700 kPa (Fig. 4e). Interestingly, the balance between the sensitivity and linear sensing range can be effectively overcome by careful design of the unit cell in the GPM. For instance, increasing the base size of the unit cell from 500 to 700 μm results in a stronger electric field with higher nonlinear changes in GPML$_{700}$ (Supplementary Fig. 6c). The much larger electric field change helps compensate for the slower increase of the capacitance at the later compression stage (higher pressure loading) to maintain linear capacitance changes. As a result, the increased sensitivity does not come at a cost of a considerably reduced linear sensing

range compared with the uniform structure (Supplementary Fig. 7a, b). As the IL concentration increases from 20 to 35 and then to 50 wt%, the sensing performance changes from 0.91 kPa$^{-1}$ over 3000 kPa to 2.95 kPa$^{-1}$ over 2000 kPa and then to 33.7 kPa$^{-1}$ over 1700 kPa (Fig. 4f). The iontronic capacitive pressure sensor with high sensitivity over ultrabroad linear sensing range compares favorably with the previous literature reports[2,9,14,17,20–23,45–56] (Fig. 5a, Supplementary Fig. 8, and Table 6).

The flexibility of the sensor allows it to easily attach to a curved surface (Supplementary Fig. 9a). The enhanced performance of the pressure sensor (e.g., high sensitivity and linearity) usually benefits from innovative structure designs. However, the sensor performance is significantly reduced under bending conditions[33] likely due to the change in the microstructures. As the bending radius is decreased from 45 to 31 and then to 13 mm, the sensitivity of GPML$_{700}$ decreases from 11.96 to 6.10 and then to 1.86 kPa$^{-1}$ (Supplementary Fig. 9b). This decrease results from the increased initial contact area and the reduced gap between the structured bottom electrode and the dielectric layer, increasing the initial capacitance (before pressure loading). The increased initial capacitance reduces the capacitance changes of the sensor under pressure and thus the sensitivity. Additionally, the nonuniform contact to the gradient pyramids upon bending reduces the sensing range from 1700 kPa to ca. 500 kPa (Supplementary Fig. 9c). However, the iontronic pressure sensor still exhibits reasonably good sensitivity in a relatively large linear sensing range due to the small initial contact area and gradually increased contact area with the increasing pressure. Further improvement can be possible when the iontronic pressure sensor is directly fabricated on the 3D freeform surfaces with a modified laser setup[57].

Besides the high sensitivity over the ultrabroad pressure range, the sensor with the gradient microstructures also exhibits a fast response/recovery of 6/11 ms from a pressure loading of 5 kPa (Fig. 5b, Supplementary Movie 2 and 3). The rapid response/recovery time is better than the existing piezoresistive[58–60], capacitive[22,45], and piezoelectric[61,62] pressure sensors, and the biological skin with a response time of 30–50 ms[63]. The initially small contact area at the fine

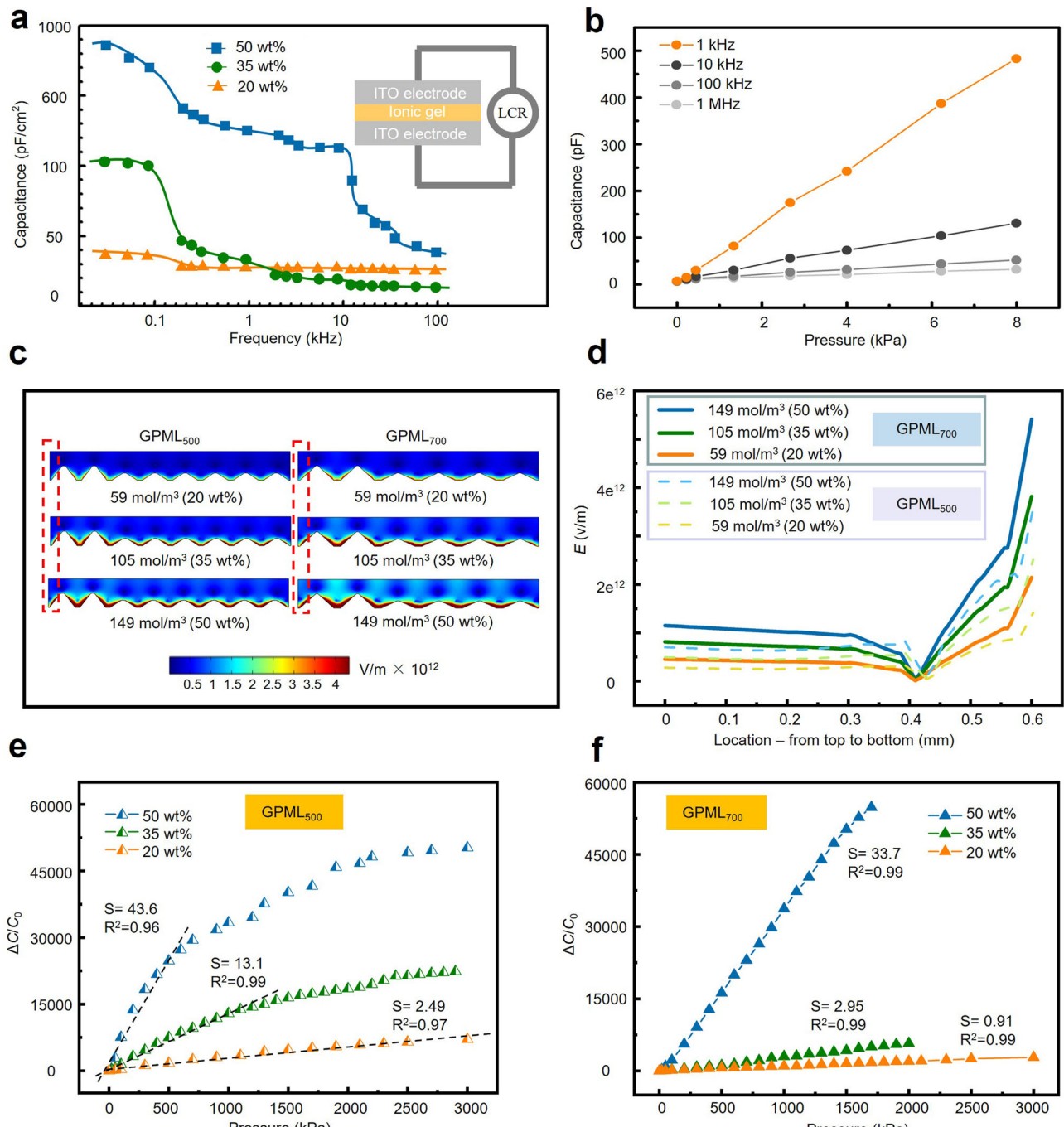

**Fig. 4 | Modulation of the electron double layer (EDL) effect for enhanced pressure sensing performance. a** Dependence of the areal capacitance on the measurement frequency for the dielectric layer with different IL concentrations (20, 30, and 50 wt%). **b** Changes in the capacitance with increasing pressure up to 8 kPa at measurement frequencies of 1 kHz, 10 kHz, 100 kHz, and 1 MHz (IL of 20 wt%). **c** Simulation of the electric field in the GPML with varying IL concentrations and the **d** highly nonlinear electric field across the space between top and microstructured bottom electrodes (in the region marked by the red dashed box in **c**). Responses of the sensors based on **e** GPML$_{500}$ and **f** GPML$_{700}$ with various weight percentages of ILs.

tip of the pyramid created from the laser-ablated rough surface (Supplementary Fig. 10a) further provides the iontronic pressure sensor with a low detection limit of 0.36 Pa to detect a light weighing paper (Fig. 5c and Supplementary Movie 4). Additionally, the capability to detect small pressure changes under high pressure (denoted as pressure resolution) is vitally important in practical applications. Even after a high pressure of 200 kPa (from a weight of 2 kg) is applied, the additional small pressure of 100 Pa, 200 Pa, and 1 kPa from a weight of 1 g, 2 g, and 10 g sequentially applied on the sensor can still be successfully captured (Fig. 5d) to give an ultrahigh pressure resolution.

The sensor response is stable and repeatable over cyclic loading at a pressure of 800 kPa (Fig. 5e), with almost negligible drift in the sensor response over 4500 cycles. The batch-to-batch variation in the sensor response is also small with a relative standard deviation (RSD) of 3.11% from three different samples (Fig. 5f), due to the highly reliable fabrication process.

## Detections of static and dynamic subtle motions
The flexible iontronic capacitive pressure sensor capable of detecting tiny pressures and subtle motions can be applied to monitor

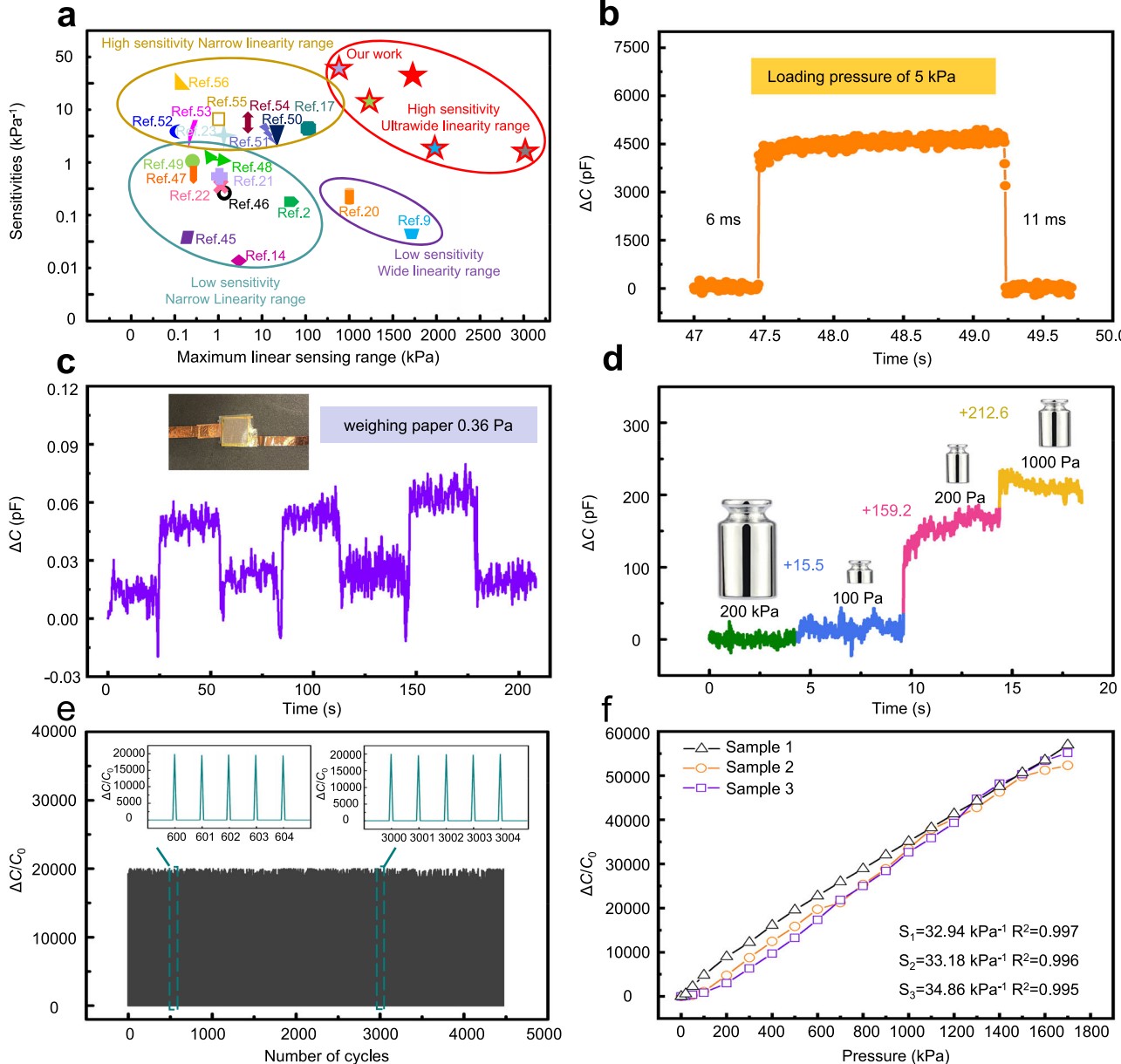

**Fig. 5 | Pressure sensing performance characterization of the iontronic pressure sensor (IL of 50 wt%). a** Performance comparison in the sensitivity and maximum linear sensing range between the flexible iontronic pressure sensor from this work and other capacitive pressure sensors. **b** Typical response curve of the iontronic pressure sensor operated at 1 kHz for the calculation of the response/ recovery time. **c** Demonstration of the detection limit of 0.36 Pa with repeated loading of a small piece of weighing paper on the sensor. **d** Detection of the additionally applied small pressure (from a weight of 1 g, 2 g, 10 g) under an existing preload pressure of 2 kg. **e** Repeatability test of the sensor to a cycling pressure load of ca. 800 kPa over 4500 cycles. **f** Response from three different pressure sensors.

physiological signals from unconventional locations. Besides the commonly demonstrated pulse measurements from the jugular vein or wrist artery[7,64–66], the far weaker pulse signal from the fingertip (ca. 10 Pa) can be detected with a clear characteristic P-, T-, and D-waves[67] by our iontronic pressure sensor (Fig. 6a). The heart rate is increased from 68 to 104 beats per minute, as the healthy human subject (28 yr, female) transitions from resting to exercise (Fig. 6b). Besides the convex surface to allow stable signal detection during motion[43,51], the fingertip pulse can also be easily accessed from smart Internet-of-Things or during human-machine collaborations. Attaching the sensor to a mouse with a double-sided tape measures actions from the single/double/triple clicks and hold-and-drag, demonstrating the reliability and stability over both static and dynamic pressure (Supplementary Fig. 10b).

## High pressure resolution under large pressure preloads

The drastically reduced (or saturated) sensitivity of most microstructured pressure sensors beyond 100 kPa limits their use to low pressure applications. Additionally, the low limit of detection often comes at a cost of a narrow sensing range and the tiny pressure detection is also only possible at nearly vanishing pressure preload (i.e., low pressure resolution). The GPM with fine features on the pyramidal surface in our iontronic pressure sensor allows it to detect tiny pressure under heavy load (or high pressure) to achieve ultrahigh pressure resolution. The representative proof-of-concept demonstration involves the use of a smart chair with four sensors (encapsulated by PMMA) placed under each leg of the chair (weight of ca. 10 kg). With an area of 1 cm² in each sensor (4 cm² in total), the human subject and the chair with a weight of up to 100 kg can generate a maximum loading

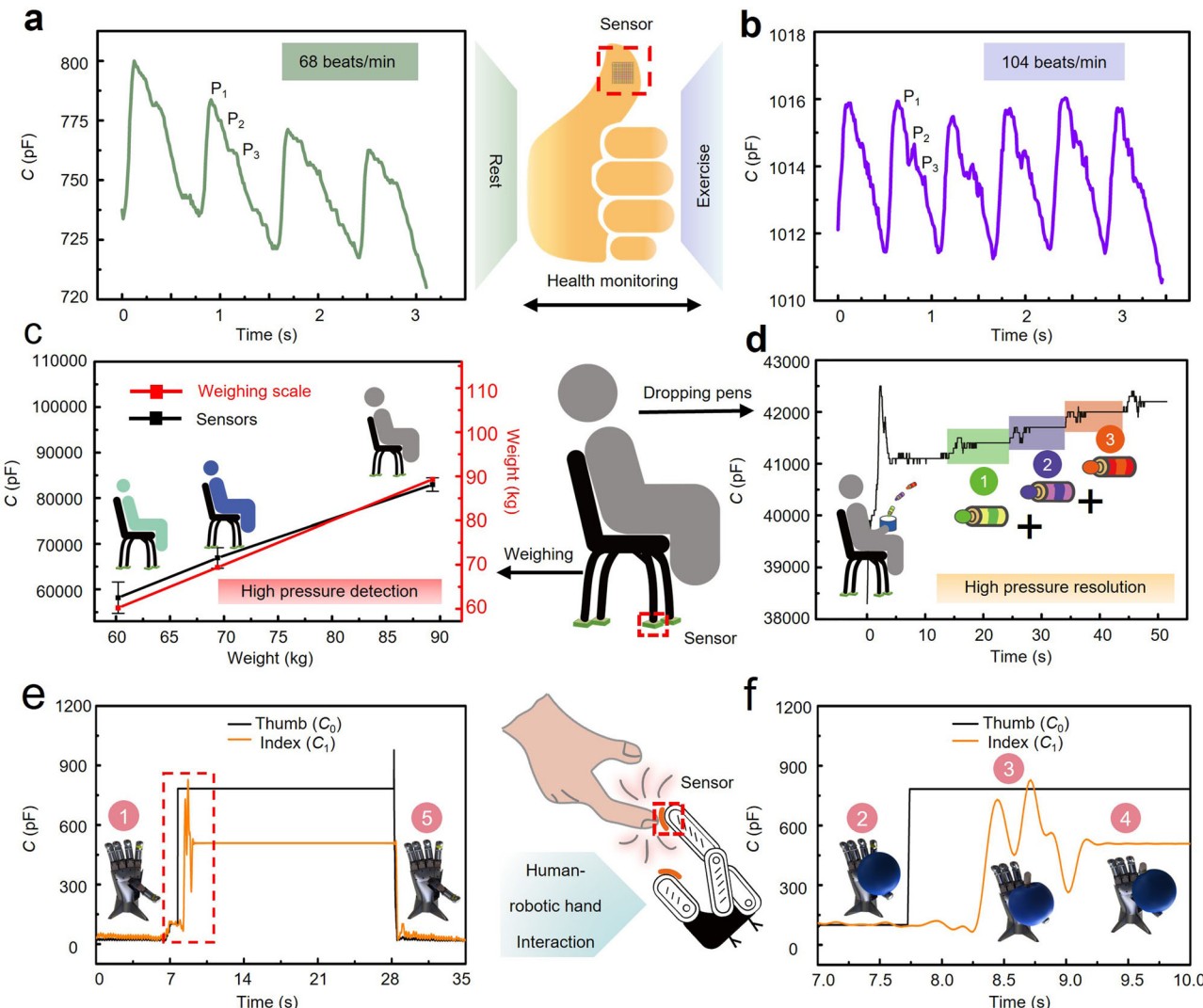

**Fig. 6 | Applications of the iontronic pressure sensor in fingertip monitoring, smart high-precision weight scale, and human-robot interaction.** Thumb pulse measurements from the healthy human subject **a** in rest and **b** after exercise. **c** Capacitive response of the sensor under the chair to the volunteer of different weights, compared to the measurements from the commercial weight scale. The error bar represents the standard deviation. **d** Demonstration of high-pressure resolution with additional tiny pressure of 145 Pa detected under a large preload of 2 MPa. **e** Capacitance responses from two sensors on the robotic fingers during handling a balloon with a zoom-in view **f** shown in the inset. (IL of 20 wt% in **a**–**d** and 50 wt% in **e**).

pressure of up to 2500 kPa on each sensor. The linear capacitive response of the sensor accurately measures the weight of three healthy human subjects (60.2, 69.4, and 89.3 kg) sitting on top, as validated by the measurements from a commercial weight scale (Fig. 6c). The additional tiny pressure under a large preload can also be accurately captured with a person (69.4 kg) holding a brush pot on the chair (preload pressure of ~2000 kPa) to catch dropping pens (each of 5.8 g, pressure of ~145 Pa) in sequence. The pronounced capacitive changes clearly capture and distinguish each dropping event (Fig. 6d and Supplementary Movie 5). The pressure resolution of 0.00725% under a preload of 2 MPa may outperform the commercial weight scale (Supplementary Fig. 10c), human skin (~7%)[68], and other microstructured pressure sensors previously reported in the literature[9,20,43,44].

### Interactive robotic hand with closed-loop control

The interactive robotic hand is demonstrated by attaching two iontronic pressure sensors (IL of 50 wt%) on the thumb and index fingers of the robotic hand (Supplementary Movie 6), with the former to detect the object and the latter to grab the object with a suitable force (Fig. 6e, f). Actuated by the rotation ($\theta$) of servo motors placed at the end of the finger joints, the fingers change vertical displacement due to

the kinematic constraint to move the robotic hand (Supplementary Fig. 11a). The dexterous manipulation of delicate objects with the robotic hand is achieved by using a proportion-integration-differentiation (PID) controller, with the flow chart shown in Supplementary Fig. 11b. Setting a threshold value ($C_b$) for the target operation prevents accidental trigger (e.g., variation/vibration in the environment) of the robotic hand (Fig. 6e, stage 1). After gently placing the balloon on the thumb, the capacitance ($C_0$) of the sensor on the thumb instantly increases to ~800 pF upon contact to exceed the preset $C_b$ (Fig. 6f, stage 2). As a result, the system identifies the object and initiates the operation of the PID controller to move the index downward for grabbing. The desired capacitance value of the sensor on the index finger can be determined or measured as $C_{1setpoint}$ for the target operation such as handling objects. The difference between the actual measured capacitance value $C_1$ from the sensor on the index finger and $C_{1setpoint}$ allows the PID controller to calculate the needed rotation output ($\Delta\theta$) in the motor to drive the index finger. The iterative process is revealed by the measured fluctuations in the $C_1$ values around $C_{1setpoint}$ (Fig. 6f, stage 3). After completion of the movement based on the feedback closed-loop control (<1 s), the balloon is gently grabbed between the thumb and index fingers with a suitable force

(Fig. 6f, stage 4). Release of the balloon from the robotic hand is then triggered by vertically pinching the balloon with a human hand, which reduces the balloon/sensor contact to a lower sensor response (either $C_1$ or $C_0$) than the preset threshold (Fig. 6e, stage 5).

## Discussion

In summary, this work reports a scalable manufacturing method to fabricate flexible iontronic pressure sensors based on programmable gradient pyramidal microstructures by leveraging the Gaussian power distribution of a $CO_2$ laser. The combination of the GPM electrode and nanoscale ultrathin IL allows the formation of EDLs at the ion gel/electrode interface, which contributes to a high sensitivity of 2.49 $kPa^{-1}$ over a broad linear sensing range of 3000 kPa. Modulation of the pyramidal profiles in the unit cell of GPM and the concentration of IL further enhances the capacitive pressure sensing performance to result in an ultrahigh sensitivity of 33.7 $kPa^{-1}$ over an ultrawide linear sensing range of 1700 kPa. Taken together with the coupled mechanical-electric simulations, the versatile modulation of these parameters in the sensor fabrication provides a viable design toolkit to potentially break the limitation in the balance between the sensitivity and linear sensing range. Additionally, the fine features on the GPM surface provide minimized initial contact area for the sensor to detect an ultralow pressure of 0.36 Pa without preload and tiny pressure variation of 145 Pa with a pressure preload of 2 MPa for an ultrahigh high pressure resolution of 0.00725%. The potential of the flexible iontronic sensor with high performance is highlighted and showcased in the proof-of-concept demonstrations of fingertip pulse monitoring from human-machine interactions, high-performance smart chairs with ultrahigh pressure resolution, and intelligent robotic hands for dexterous object manipulation. The represented demonstrations presented in this study open up opportunities for future wearable health-monitoring devices, smart prosthesis skins, and intelligent human-machine collaboration systems.

Despite the high sensitivity in a broad linear sensing range, further performance improvements are possible. For example, the sensitivity can be enhanced by increasing the concentration of ILs and the sensing range can be extended by using larger pyramids for more significant contact area changes. The detection limit can be further decreased by including an air gap between the electrode and the dielectric layer. The linear sensing range can also be extended by adjusting the microstructures due to their role to modulate the capacitance changes. Further optimization of the laser parameters to create various height combinations in the microstructures and the theoretical analysis to account for bending and to predict the other sensing parameters will be pursued in our future works.

## Methods

### Materials and chemicals

Silicone rubber (PDMS, Sylgard 184) was purchased from Dow Chemical Co., Ltd. PET/ITO conductive film, (3-Aminopropyl)triethoxysilane (APTES), and PVDF-HFP (with a molar mass of $M_w = 400,000$ g $mol^{-1}$ and $M_n = 130,000$ g $mol^{-1}$, respectively) were obtained from Sigma-Aldrich. Acetone was obtained from VWR Chemicals BDH Co., Ltd. The ionic liquid 1-ethyl-3-methylimidazolium bis(tri-fluoromethylsulfonyl) imide ([EMI][TFSI]) was purchased from Fisher Scientific Co., Ltd. PMMA with a thickness of 2 mm was purchased from Zonon@Amazon. The robotic hand was purchased from Youbionic Co., Ltd.

### Fabrication of the PMMA mold

The PMMA mold was created by laser scribing with the $CO_2$ laser (VLS 2.30 universal laser system (30 W, wavelength of 10.6 μm, Inc. VLS2.3) under ambient conditions with the patterns created in Inkscape. The gradient structure was fabricated using raster mode with 20% of the maximum speed, varying power between 0–30% of the maximum power, 1000 PPI, and an image density of 6. The gradient base of the

microstructures with a reduced height from outside to inside was first formed by gradually decreasing laser power from the black to red and then to the purple region (Supplementary Fig. 1). Next, the reduced laser power was used in the subsequent laser processing step at the same location to form pyramidal structures (Supplementary Table 3). The comparison between microstructures obtained from two PMMA templates that were separately created by using the same laser parameters demonstrated the reasonably good consistency of laser fabrication (Supplementary Fig. 12).

### Preparation of the bottom electrode

The mixed base and curing agent of Sylgard 184 with a weight ratio of 10:1 was spin coated onto the PMMA mold at a speed of 400 rpm to obtain a uniform layer with a thickness of ca. 250 μm. After degassing and curing in the oven at 90 °C for 1 h, slowly peeling the cured PDMS layer off from the PMMA mold was followed by air plasma (PDC-001-HP, Harrick Plasma) treatment at 30 W for 10 min. Next, the PDMS layer was dipped in the APTES solution (concentration of 15 mM with 35.1 μL in 10 mL DI water) for 12 h to achieve strong adhesion followed by sputtering of 50-nm-thick Au (Quorum Q150R, Sputter Coater) as the bottom electrode.

### Preparations of the top electrode and iontronic layer

After preparing the acetone and PVDF-HFP with a mass ratio of 20:3, the IL [EMI][TFSI] was added with a mass ratio of 20%, 35%, or 50% of PVDF-HFP to obtain the mixture. The mixture was heated and stirred at 120 °C and 1000 rpm until PVDF-HFP was completely dissolved to obtain the uniform solution. The solution was then spin-coated on the as-received ITO/PET film (cut to 1.2 cm × 1.2 cm) at 1000 rpm for 30 s and dried on the hot plate at 80 °C in a fume hood to fully evaporate the acetone and cure the iontronic layer.

### Pressure sensor assembly

A transparent medical tape (3 M, Tegaderm™ film) was used at the edge between the bottom and top electrodes to assemble the pressure sensor and also used to tightly attach the sensor to the finger for pulse detection at the fingertip. Four pressure sensors sandwiched between two PMMA boards (1.5 cm × 1.5 cm × 0.1 cm) were placed under each leg of the chair for weight measurements and demonstrations of high pressure resolution. The as-packaged pressure sensors were used for other applications unless otherwise specified.

### Characterization and measurements

The surface morphology of the bottom electrode was characterized by field emission-scanning electron microscopy (Zeiss FESEM Gemini 500). The capacitance of the pressure sensor (10 mm × 10 mm) was measured by an LCR meter (Hioki IM3536) at a frequency of 1 kHz, unless specified otherwise. A force gauge (HLB Test Stand+HP-500N, Mxmoonfree Co., Ltd) was used to apply and record the pressure. Signals from two capacitors for robotic hand application were measured by the Arduino nano board[69]. The repeatability test was performed by using a linear rail guide slide actuator. Areal capacitances were measured by sandwiching the ionic gel between two flat PET/ITO electrodes. Smoothening of the noisy data (stage 3 in Fig. 6f) was achieved by applying an FFT filter with 5 points of window in Origin.

### Finite element analysis

The mechanical and electrical finite element analysis (FEA) was performed using the commercial package COMSOL Multiphysics 5.6. In the mechanical simulation, the structured bottom PDMS electrode was modeled as an incompressible Mooney-Rivlin material, with Young's modulus of 750 kPa and Poisson's ratio of 0.49. A pressure of 1600 kPa was applied on the top electrode that was treated as a rigid plate (not shown in Fig. 3a). The initial electrode/dielectric contact interface ($A_0$) is determined at the pressure of 5 kPa. In the coupled EDL and

mechanical simulation, the distribution of the electric field in the iontronic layer was obtained by applying pressure and an electric potential of 3 V on the top ITO electrode and grounding the bottom gold. After merging PVDF and KOH as the iontronic medium with a diffusion coefficient of $1e^{-9}$ m²/s, the initial relative concentration of the iontronic medium was set to 0 for computational efficiency without the loss of generality. The concentration of the iontronic layer at the gold boundary was set as $Con$ mol/m³ with respect to the initial concentration. The electric field distribution was computed by coupling mechanical, electrostatics, and transport of dilute species modules (Supplementary Note 3). The transport of dilute species was driven by the generated electric field from the applied potential on the top electrode and the generated polarization field due to the pressure applied on piezoelectric PVDF. On the other hand, the transport of the diluted species generated an electric field due to the space charge density coupling. Increasing the pressure and concentration at the ITO electrode resulted in the increased electric field distribution, which further facilitated the transport of the ions in the iontronic medium. It is worth noting that time-varying pressure may contribute to a time-varying electric displacement field, which may result in time-varying free surface charge density for an increased rate of transport of ions.

### Reporting summary

Further information on research design is available in the Nature Portfolio Reporting Summary linked to this article.

## Data availability

All data that support this study are available within the article and its Supplementary Information. Other relevant data are available from the corresponding authors upon request.

## Code availability

The codes generated in this study are available from the corresponding authors upon request.

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

## Acknowledgements

H.C. acknowledges the support provided by NIH (Award Nos. R21EB030140, U01DA056242, R33HL154215, and R21OH012220), NSF (Grant Nos. 1933072 and 2222654), and Penn State University. R.Y. acknowledges the support provided by the Joint Doctoral Training

Foundation of the Hebei University of Technology. The insightful discussion with Xue Chen at the Hebei University of Technology, Honglei Zhou at the Tsinghua University, and Zhoulyu Rao at the Pennsylvania State University is also acknowledged.

## Author contributions

R.Y. and H.C. conceived the ideas and designed the research. A.D. performed the modeling and simulations. B.L. prepared the electrodes. N.T. participated in the discussion of the results. W.Z. conducted the programming. Z.N. participated in the design of figures. Y.G. and X.X. helped with the sensor testing and discussed the results. D.E. performed SEM. R.Y. and H.C. wrote the manuscript with editing from T.L. and input from all authors.

## Competing interests

R.Y. and H.C. are filing a provisional patent on this work. The remaining authors declare no competing interests.
