## [Peer Review File · Nature Communications]

Iontronic pressure sensor with high sensitivity over ultra-broad linear range enabled by laser-induced gradient micro-pyramidsREVIEWER COMMENTS

Reviewer #1 (Remarks to the Author):

In this work, the author presented a facile and low-cost fabrication method to prepare the iontronic pressure sensor with super-high sensitivity and a wide linear sensing range. The author also showed the potential of the prepared sensor in practical application. This is an interesting work and would be appealed in broad communities, I will recommend this work to publish in nature communications after addressing the following issues.

1. Does the height of the pyramid structure have any effect on the performance of the as-prepared iontronic pressure sensor?
2. The author should show the sensitivity of the sensor under different bending conditions.
3. The references should be more concise.

Reviewer #2 (Remarks to the Author):

Manuscript NCOMMS-22-32774-T entitled "Iontronic pressure sensor with ultra- broad linear range and high sensitivity enabled by laser-induced gradient pyramidal structures" by Yang et al., reports on the design and fabrication of a pressure sensor that displays a linear response to pressure up to 1500 kPa while preserving a high sensitivity of ca. 33 kPa⁻¹ through the use of pyramid micro structures of different height. The approach is novel, the results are timely and represent a significant contribution to the field. I will support publication of this manuscript after the following deficiencies are corrected.

Statistics and analysis.

- 1) The manuscripts lacks statistical data to give the reader an opportunity to get a sense of how large sample-to-sample variations are. Despite claims in line 275, Figure 5f suggest that the sensitivity and linearity can vary significantly between samples, authors should present data analysis and let the reader decide if variations are truly negligible.
- 2) In relation to comment 1) and the rest of analysis presented with respect to the linearity of the response, it is important that the authors clarify the pressure range under which R² was evaluated. For instance, in Fig 3d, data for the GPML500 is labeled as having an R² of 0.971 up to 3000 kPa, but data from sample GPMS500 appears to display a more linear response up to 2000 kPa while labeled as having an R² of 0.952. Another open question that would need some supporting data (comment 1) is if these variations are significant or just due to device-to-device variations.

Low-cost claim.

- 3) The authors overemphasize that their sensors are low cost, yet this is rather subjective metric. Particularly when they use sputtered gold, the device yield is unknown and it is unclear what number is used to gauge the claim of low cost. Since cost assessments for lab samples is rather subjective and fairly irrelevant to what the technology will cost to produce in an industrial setting and for how much it could sell in the market, I would strongly suggest that the authors preferably drop, or make this claim quantitative (i.e. estimate how much on of their sensors would cost?). In my view, this claim is unimportant to support the novelty of the approach and importance of results.

Qualitative assessments

- 4) Authors should avoid making qualitative and unsupported assessments. For instance, in line 100 the authors claim that their work outperforms previous reports without citing specific references, by how much or in which aspects. Instead, they refer the reader to Figure b, which is a qualitative comparison that is not supported by any data in the manuscript or with relevant references. Fig. b needs to be supported by a table with values used and references. Other instances of qualitative assessment in the use of adjectives exist in the paper and should be corrected by making an effort to provide quantitative assessments, e.g. line 211 use of "stronger" or "dramatically increase" should be replaced by something like: "...increased ion mobility by X leading to a Y times increase in the areal capacitance..."

Other deficiencies

5) Layers in Figure 1a should be labeled.

6) Figure 4 presents data with IL concentration and simulations based on IL molarities, making it difficult for the reader to relate on data set with the other. Authors should point out and label which IL concentrations correspond to which IL molarities on the graphs.

7) Line 401, in the methods authors talk about black, red and green regions without referring to any figure, making it impossible to understand what they have done or what regions they refer to. Please explain so that an interested reader may reproduce the results.

Reviewer #3 (Remarks to the Author):

The manuscript entitled "Iontronic pressure sensor with ultra-board linear range and high sensitivity enabled by laser-induced gradient pyramidal structures" introduced a flexible pressure sensing structure with high linearity and sensitivity using a gradient microstructure. This concept is not new (similar to Ref 34), the results are not better than that of the prior arts, and the intro is lack of many key references in the field of iontronic sensing. Therefore, the reviewer doesn't believe this manuscript reaches the quality of publication of Nature Communications. Several detailed comments have been included below.

1. The authors claimed that the sensor presented in this paper possesses the advantages of low fabrication cost. However, according to Figure S1, the fabrication of the gradient-structured electrode is extremely complicated and time-costed, so the low fabrication cost cannot be the major advantage of this sensor.

2. In Figure 1b, the authors announced that conventional EDL-based sensor is poor at linearity. However, many papers (Nature Communications, 2020, 11,209; Adv. Funct. Mater. 2019, 1807343) have proved that the iontronic pressure sensor can also present high linearity even better than conventional capacitive sensors.

3. Figure 1a is confusing, and we still don't understand the detailed structure of the sensor according to this image.

4. Most importantly, a theoretical model should be studied to guide the structure design. Why the gradient microstructure could achieve a higher linearity? A theoretical equation is necessary. Current study only focus on the phenomenon.

5. The pyramidal structures are too rough, a finer method should be used to fabricate the structure.

6. The sensitivity of the sensor is highly related with the initial capacitance of the sensor. The initial capacitances of the sensors in Figure 2d and 2e should be informed. If the high sensitivity of the sensor derived from the extremely low initial capacitance? If yes, why the sensor has such low initial capacitance?

7. In figure 3, how to determine the initial contact area A_0 of different structures?

8. In figure 5b, finger touch is not a quantitative method to evaluate the response time. Please using the standard method described in other papers.

9. In figure 5c, we still cannot judge if the fluctuation of the curve comes from the noise or the tissue.

10. If the high linearity of the sensor can be retained when bending? Seems that the high linearity cannot be achieved when the pressure is applied non-uniformly.

Reviewer #4 (Remarks to the Author):

Manuscript titled " Iontronic pressure sensor with ultra-board linear range and high sensitivity enabled by laser-induced gradient pyramidal structures" is a nice work related to Development and testing of system for pressure sensing application.

Recommendation: Review Again After Resubmission (Paper is not acceptable in its current form, but has merit. A major rewrite is required. Author should be encouraged to resubmit a rewritten version after the changes suggested in the Comments section have been completed.)

Before publishing, the authors must revise the paper as per comments mentioned below:

1. Authors have developed and tested a system for pressure sensing application which is of good quality. The work supports the conclusions and claims. Data analysis, interpretation and conclusions presentation is satisfactory. Methodology is fine.
2. The work is significant and can be considered for publication in the journal after some improvements as suggested next.
3. Is there enough detail provided in the methods for the work to be reproduced? Sensing range of 1700 kPa is quite good. What about its Low and High Limits? Can the low limit be decreased further in future?
4. "The development of capacitive pressure sensors with high sensitivity in a wide linear range remains elusive due to the trade-off between sensitivity and linear range." Add some quantitative data from published literature here to support this statement. Last paragraph of Introduction Section should highlight research gaps, and main objectives of this work.
5. Restructuring of the paper is required for easy readability
6. Fig. 1 shows the methodology. Authors are advised to bifurcate Section 2 into Two Major Sections, that is, Section 2: Materials and Methods. Section 3: Results and Discussion. Fig. 1 can then be good for Section 2: Materials and Methods
7. FEA is good.
8. Figure 3a: Mpa should be MPa
9. Section 4.4: the authors should mention that how much displacement is applied on the top electrode? Is it a pressure load? If it is so, please mention the magnitude.
10. The authors should compare the sensitivities of different laser-induced structures used for the simulations, rather than just mentioning it for GPML only.
11. "The electric field distribution was computed by coupling mechanical, electrostatics, and transport of dilute species modules." Authors may add the governing mathematical model behind this coupling process and then it can be easily verified using some other standard software also such as MATLAB.
12. "Increasing the pressure and concentration at ITO electrode resulted in the increased electric field distribution, which further facilitated the transport of the ions in the iontronic medium." Does increase in pressure only facilitates or increases the rate of transport of ions also?

December 14, 2022

Reviewer #1:

In this work, the author presented a facile and low-cost fabrication method to prepare the iontronic pressure sensor with super-high sensitivity and a wide linear sensing range. The author also showed the potential of the prepared sensor in practical application. This is an interesting work and would be appealed in broad communities, I will recommend this work to publish in nature communications after addressing the following issues.

Our response: We highly appreciate the reviewer's positive evaluation of our work. We also appreciate the insightful comments that help significantly improve the overall quality of this work.

1. *Does the height of the pyramid structure have any effect on the performance of the as-prepared iontronic pressure sensor?*

Our response: We thank the reviewer for this question. The pyramid structures with excessive height (e.g., over 1 mm) can be difficult to encapsulate or package. Buckling may also occur for those with a high aspect ratio as discussed in 2nd paragraph in the “**Gradient pyramidal microstructures from laser-ablated molds**” Section and Supplementary Fig. 3b “As a result, the nonlinear deformation from buckling^{1,2,3} compromises the sensor performance in terms of linearity.”

The pressure-sensing range is directly relevant to the deformation of the microstructure. In general, the microstructures with high aspect ratios (or larger size) are beneficial for the increased sensitivity⁴ (or increased sensing range). As reported in the literature, the pressure sensors with a microstructure size smaller than 100 μm often show a sensing range of less than 50 kPa (Supplementary Table 4). In contrast, the sensors

with a large microstructure size (e.g., side length bigger than 500 μm) can provide a sensing range of more than 1000 kPa^{5,6}. As for the pyramidal microstructures, when the L/H ratio (L is the bottom side length and H is the height) is $\sqrt{2}$, the sensors can exhibit a balanced performance between sensitivity and linearity^{7,8,9}.

To imitate pyramid structures and also avoid buckling, we design the gradient structure GPML₇₀₀ structure with L/H of 1.2 and 2.2, which resulted in linear sensing ranges (for all three ionic liquid concentrations).

As a proof-of-the-concept demonstration, the gradient structure prepared with the simple fabrication method from this work outperformed the other pressure sensors reported in the literature (Supplementary Table 4). Further optimization of the laser parameters to create various height combinations in the microstructures may help modulate the sensor performance, which will be pursued in our future studies.

Supplementary Table 4: Microstructures with different aspect ratios used in the pressure sensors

Principle	Structure	Side length (L) (μm)	Height (H) (μm)	L/H	Maximum Sensing range (kPa)	Ref.
Piezoresistant	Pyramid	4.64	2.97	1.56	3	10
Conventional capacitor	Pyramid	50	30.25	1.65	35	11
Conventional capacitor	Pyramid	4.88	1.65	2.97	7	12
EDL-based capacitor	Pyramid	6.49	3.5	1.85	50	13
Piezoelectric	Pyramid	60	42	1.42	10	14
Conventional capacitor	Gradient Dome	500	700; 450; 200	/	1700	5

EDL-based capacitor	Gradient pyramid	700	570; 310	1.2; 2.2	3000	
------------------	-----	----------	----------	------	--

Our modification to the manuscript: (1) We have added a new paragraph in the “Discussion” Section for future studies.

“The represented demonstrations presented in this study open up opportunities for future wearable health-monitoring devices, smart prosthesis skins, and intelligent human-machine collaboration systems.

Despite the high sensitivity in a broad linear sensing range, further performance improvements are possible. For example, the sensitivity can be enhanced by increasing the concentration of ILs and the sensing range can be extended by using larger pyramids for more significant contact area changes. The detection limit can be further decreased by including an air gap between the electrode and the dielectric layer. Further optimization of the laser parameters to create various height combinations in the microstructures may help modulate the sensor performance, which will be pursued in our future studies.”

(2) We have included the above-mentioned explanation for designing pyramids in Supplementary information (Note 1 and Table 4). We also updated the manuscript.

“By modulating the sizes of the laser pattern while maintaining the ratio among three layers, different sizes of pyramidal microstructures can be formed (Supplementary Fig. 4de). Different heights in the resulting pyramidal microstructure can also be achieved by including an additional layer created from varying powers before laser ablating the tri-layered structures in the PMMA template (Supplementary Fig. 5a-c; see details in Supplementary Note 1 and Table 4).”

2. The author should show the sensitivity of the sensor under different bending conditions.

Our response: Many thanks for the reviewer's suggestion. We have added a new paragraph in the "Sensing performance modulated by the enhanced EDL effect" Section and a corresponding plot (Supplementary Fig. 9b) to discuss the sensitivity of the sensor under different bending conditions.

Our modification to the manuscript:

Supplementary Fig. 9. b The pressure sensing performance of the iontronic pressure sensor under different bending conditions.

The flexibility of the sensor allows it to easily attach to a curved surface (Supplementary Figure 9a). The outstanding performance of the pressure sensor (e.g., high sensitivity and linearity) usually benefits from ingenious structure designs. However, the sensor performance is often significantly reduced under bending conditions¹⁵ likely due to the change in the microstructures. As the bending radius is decreased from 45 to 31 and then to 13 mm, the sensitivity of GPML₇₀₀ decreases from 11.96 to 6.10 and then to 1.86 kPa⁻¹ (Supplementary Fig. 9b). This decrease results from the deformation in the highest pyramid and the reduced gap between the structured bottom electrode and the dielectric layer, increasing the initial capacitance (before pressure loading). The increased initial capacitance reduces the capacitance changes of the sensor under pressure and thus the sensitivity. Additionally, the

bending strain also causes an uneven pressure loading on lower pyramids due to deformed pyramids, greatly reducing the linear sensing range to ca. 500 kPa compared with 1700 kPa before bending. However, the iontronic pressure sensor still exhibits reasonably good sensitivity in a relatively large linear sensing range. Further improvement can be possible when the iontronic pressure sensor is directly fabricated on the 3D freeform surfaces with a modified laser setup¹⁶.

3. *The references should be more concise.*

Our response: Many thanks for the reviewer's careful check. We have updated the format of references.

Our modification to the manuscript: The format of references has been updated per journal guidelines.

Reference:

1. Yang W, *et al.* Multifunctional Soft Robotic Finger Based on a Nanoscale Flexible Temperature–Pressure Tactile Sensor for Material Recognition. *ACS Applied Materials & Interfaces* **13**, 55756-55765 (2021).
-

Reviewer #2:

Manuscript NCOMMS-22-32774-T entitled "Iontronic pressure sensor with ultra- broad linear range and high sensitivity enabled by laser-induced gradient pyramidal structures" by Yang et al., reports on the design and fabrication of a pressure sensor that displays a linear response to pressure up to 1500 kPa while preserving a high sensitivity of ca. 33 kPa-1 through the use of pyramid micro structures of different height. The approach is novel, the results are timely and represent a significant contribution to the field. I will support publication of this manuscript after the following deficiencies are corrected.

Our response: We appreciate the referee's highly positive evaluation of this work. Many thanks for the reviewer's comments and suggestions that help significantly improve the overall quality of this work.

Statistics and analysis.

1) The manuscripts lacks statistical data to give the reader an opportunity to get a sense of how large sample-to-sample variations are. Despite claims in line 275, Figure 5f suggest that the sensitivity and linearity can vary significantly between samples, authors should present data analysis and let the reader decide if variations are truly negligible.

Our response: Many thanks for the reviewer's suggestion. We have updated Fig. 5f and provided the sensitivity and linearity of each sample.

Our modification to the manuscript:

(1) We have updated Fig. 5f and provided the sensitivity and linearity of each sample. We have also calculated the relative standard deviation (RSD) and added it to the manuscript.

“The batch-to-batch variation in the sensor response is also small with a relative standard deviation (RSD) of 3.11% from three different samples (Fig. 5f), due to the highly reliable fabrication process.”

(2) We have also updated one of the curves (GPML₇₀₀ 50 wt%) in Fig. 2g and Fig. 4f by using the average sensitivity from the updated Fig. 5f.

(3) The best sensitivity (33 kPa^{-1}) has been updated to 33.7 kPa^{-1} in the manuscript.

(i) The resulting optimized sensor exhibits a high sensitivity of 33.7 kPa^{-1} over a linear sensing range of 1700 kPa, an ultralow detection limit of 0.36 Pa, and a pressure resolution of 0.00725% under ultrahigh pressure of 2000 kPa.

(ii) Combining the programmable microstructures with an ultrathin IL layers results in an iontronic sensor to break the trade-off between sensitivity and linear sensing range,

exhibiting a high sensitivity (33.7 kPa^{-1}) and excellent linearity (0.99) over an ultra-broad linear sensing range up to 1700 kPa.

(iii) As the IL concentration increases from 20 to 35 and then to 50 wt%, the sensing performance changes from 0.91 kPa^{-1} over 3000 kPa to 2.95 kPa^{-1} over 2000 kPa and then to 33.7 kPa^{-1} over 1700 kPa (Fig. 4f).

(iv) Modulation of the pyramidal profiles in the unit cell of GPM and the concentration of IL further enhances the capacitive pressure sensing performance to result in an ultrahigh sensitivity of 33.7 kPa^{-1} over an ultrawide linear sensing range of 1700 kPa.

2) In relation to comment 1) and the rest of analysis presented with respect to the linearity of the response, it is important that the authors clarify the pressure range under which R^2 was evaluated. For instance, in Fig 3d, data for the GPML500 is labeled as having an R^2 of 0.971 up to 3000 kPa, but data from sample GPMS500 appears to display a more linear response up to 2000 kPa while labeled as having an R^2 of 0.952. Another open question that would need some supporting data (comment 1) is if these variations are significant or just due to device-to-device variations.

Our response: Many thanks for the reviewer's question. The three curves in Fig. 3d are provided individually in the following, with the R^2 value added for the same pressure range from 0 to 3000 kPa. Despite the minor difference, they all exhibit very good linearity, so the focus of Fig. 3d is to compare the significant difference in the sensitivity.

In the repeatability test, Fig. 5f was updated to show the small variation among different samples.

Our modification to the manuscript:

“The batch-to-batch variation in the sensor response is also small with a relative standard deviation (RSD) of 3.11% from three different samples (Fig.5f), due to the highly reliable fabrication process.”

Low-cost claim.

3) The authors overemphasize that their sensors are low cost, yet this is rather subjective metric. Particularly when they use sputtered gold, the device yield is unknown and it is unclear what number is used to gauge the claim of low cost. Since cost assessments for lab samples is rather subjective and fairly irrelevant to what the technology will cost to produce in an industrial setting and for how much it could sell in the market, I would strongly suggest that the authors preferably drop, or make this claim quantitative (i.e. estimate how much on of their sensors would cost?). In my view, this claim is unimportant to support the novelty of the approach and importance of results.

Our response: We thank the reviewer’s suggestion. As the microstructure plays a vital role in the performance of pressure sensors, we tried to focus on the fabrication of pressure-sensitive microstructures. Compared with the commonly used method of photolithography for fabricating the designed microstructures, our method based on the cheap PMMA board and the CO₂ laser system (common in the machine shop) presents a low-cost fabrication method (Supplementary Table 1).

Supplementary Table 1. Cost comparison between the wet etched silicon wafer and laser-ablated PMMA for fabricating the microstructures

Method	Materials	Facilities	Environment requirement	Time cost
Wet etching	Silicon wafer (4.0") \$13.9/piece, Photoresist, Silicon nitride; Potassium hydroxide; Trichloro- (1H,1H,2H,2H- perfluorooctyl) silane; IPA; Acetone;	Clean room, Lithography equipment, Spin coater, wet bench, O ₂ plasma;	High temperature	> 5 h ¹⁷
Our method	PMMA board (4.0") \$0.5/piece	CO ₂ laser system	Ambient conditions	10 min

Note: The costs for materials such as silicon wafers and PMMA boards are based on catalog prices (e.g., Amazon).

Our modification to the manuscript: We have updated the manuscript to emphasize the fabrication cost of the microstructures in the abstract and the “Fabrication and working mechanism” Section.

(1) “Here we present a facile fabrication method to integrate an ultrathin ionic layer and low-cost programmable gradient microstructures.”

(2) The superior performance of the iontronic pressure sensor relies on a low-cost gradient pyramidal structure of laser-ablated polymethyl methacrylate (PMMA) that can be created by a multi-step process (Supplementary Table 1).

Qualitative assessments

4) *Authors should avoid making qualitative and unsupported assessments. For instance, in line 100 the authors claim that their work outperforms previous reports without citing specific references, by how much or in which aspects. Instead, they refer the reader to Figure b, which is a qualitative comparison that is not supported by any data in the manuscript or with relevant references. Fig. b needs to be supported by a table with values used and references. Other instances of qualitative assessment in the use of adjectives exist in the paper and should be corrected by making an effort to provide quantitative assessments, e.g. line 211 use of “stronger” or “dramatically increase” should be replaced by something like: “...increased ion mobility by X leading to a Y times increase in the areal capacitance...”*

Our response: We thank the reviewer for the careful check. (1) We have updated the description of Fig.4a. (2) We have added Supplementary Table 2 to support Fig. 1b. (3) The performance comparison in Fig. 1b involves structure fabrication cost, the detection limit, pressure resolution, response time, and linear sensing factor.

Our modification to the manuscript: (1) We have updated the description of Fig. 4a.

“Increasing the IL concentration from 20 to 50 wt% results in an increased areal capacitance from 36.5 to 984 pF cm⁻² at 100 Hz (Fig. 4a).”

(2) We have included a table (Supplementary Table 2) to support Fig. 1b.

“Taken together with the fast response/recovery time, the flexible iontronic pressure sensor from this work outperforms both the microstructured conventional and EDL-based capacitive pressure sensors in the previous literature reports (Fig. 1b and Supplementary Table 2).”

Supplementary Table 2. Performance comparison of different capacitive pressure sensors.

	Number	Type	Structure fabrication method	Sensitivity (kPa ⁻¹)	Linear Sensing Rang (kPa)	Linear sensing factor (S_p)	Pressure resolution	Response/recovery time (ms)	LOD (Pa)	Ref.
1	1	EDL: mold-based	Photolithography Silicon wafer	1.3	3	3.9	0.02% (base pressure of 5 kPa)	15/15	0.2	1
	2	EDL: mold-based	Photolithography Silicon wafer	7.49	6	44.94	NR	9/9	0.9	18
	3	EDL: mold-based	PTFE template	7.7	7.4	56.98	NR	NR	NR	19
	4	EDL: mold-based	3D print mold	49.1	4-485	2.37×10^4	NR	0.61/3.63	NR	20
2	5	EDL: structure-transfer	Transfer sandpaper structure	3302	10	3.302×10^4	0.0056% (base pressure of 320 kPa)	9/18	0.08	3
	6	EDL: structure-transfer	Transfer sandpaper structure	5.5	30	165	NR	70.4/92.8	2	21
	7	EDL: structure-transfer	Transfer sandpaper structure	9.17	0.013-2063	1.89×10^4	NR	5/16	13	22

	8	EDL: structure-transfer	Transfer sandpaper structure	6.94	100	694	NR	48/NR	2.88	23
	9	EDL: structure-transfer	Transfer sandpaper structure	200	60	1.2×10^4	1.4% (base weight of 71 kg)	98/70	20	24
	10	EDL: structure-transfer	Directly using fabric materials	6.5	10	65	NR	30/30	7.5	25
	11	EDL: structure-transfer	Directly using fabric materials	0.24	70	16.8	NR	18	35	26
3	12	Conventional: mold-based	Photolithography Silicon wafer	2.51	10	25.1	NR	84/117	2	4
	13	Conventional: mold-based	Photolithography Silicon wafer	0.022	5	0.11	NR	NR	NR	27
	14	Conventional: mold-based	Photolithography Silicon wafer	0.43	1	0.43	NR	33/33	3.4	28
	15	Conventional: mold-based	Photolithography Silicon wafer	7.68	0.5	3.84	NR	30/28	1 mg	29
	16	Conventional: mold-based	Commercially anodic aluminum oxide template	0.35	2	0.7	NR	48/60	4	30

	17	Conventional: mold-based	Commercially anodic aluminum oxide template	6.583	0.1	0.6583	NR	48/36	3	31
	18	Conventional: mold-based	Polyurethane sponge skeletons	0.062	0.3	18.6	NR	45/83	3	32
	19	Conventional: mold-based	Nickel foam template	3.13	1	3.13	NR	94/	0.07	33
	20	Conventional: mold-based	Micro-engraving plastic template	0.065	1700	111	1% (base pressure of 1000 kPa)	100/100	/	5
4	21	Conventional: structure-transfer	Transfer the outline of polystyrene spheres	30.2	0.13	3.926	NR	25/50	0.7	34
	22	Conventional: structure-transfer	Transfer the outline of polystyrene spheres	6.61	0.11	0.73	NR	100/100	1	35
	23	Conventional: structure-transfer	Transfer the outline of reed leaves	0.6	1	0.6	NR	180/120	4.5	36

	24	Conventional: structure-transfer	Transfer the outline of bamboo leaves	2.08	1	2.08	NR	500/700	20	37
	25	Conventional: structure-transfer	Transfer the outline of lotus leaves	1.2	2	2.4	NR	36/58	0.8	38
	26	Conventional: structure-transfer	Transfer the surface structure of obscured glass	1.1	0.5	0.55	NR	1000/NR	1	39
	27	Conventional: structure-transfer	Directly using dried flower petal as dielectric layer	1.54	1	1.54	NR	NR	0.6	40
	28	Conventional: structure-transfer	Transfer the surface structure of paper	0.62	2	1.24	NR	NR	6	41
		EDL: mold-based	Laser-ablated PMMA board	33.7	1700	5.729×10^4	0.00725% (base pressure of 2000 kPa)	6/11	0.36	

Note: NR (not reported, $S_P = S \cdot \Delta P$ (S : sensitivity and ΔP : the corresponding linear sensing range))

(3) We have updated Fig. 1b based on the data in Supplementary Table 2.

5) Layers in Figure 1a should be labeled.

Our response: We thank the reviewer for the careful check. We have updated Fig. 1a.

Our modification to the manuscript: We have updated Fig. 1a.

6) Figure 4 presents data with IL concentration and simulations based on IL molarities, making it difficult for the reader to relate on data set with the other. Authors should point out and label which IL concentrations correspond to which IL molarities on the graphs.

Our response: We thank the reviewer for the careful check, and we have updated Fig. 4c.

Our modification to the manuscript: The corresponding IL concentrations have been added in Fig.4c.

7) Line 401, in the methods authors talk about black, red and green regions without referring to any figure, making it impossible to understand what they have done or what regions they refer to. Please explain so that an interested reader may reproduce the results.

Our response: We thank the reviewer for the careful check and we have added the reference to the figure/table.

Our modification to the manuscript: We have updated the discussion with proper references to the figure/table.

“The gradient base of the microstructures with a reduced height from outside to inside was first formed by gradually decreasing laser power from the black to red and then to the purple region (Supplementary Fig. 1). Next, the reduced laser power was used in the subsequent laser processing step at the same location to form pyramidal structures (Supplementary Table 3).”

Reviewer #3:

The manuscript entitled “Iontronic pressure sensor with ultra-broad linear range and high sensitivity enabled by laser-induced gradient pyramidal structures” introduced a flexible pressure sensing structure with high linearity and sensitivity using a gradient microstructure. This concept is not new (similar to Ref 34), the results are not better than that of the prior arts, and the intro is lack of many key references in the field of iontronic sensing. Therefore, the reviewer doesn’t believe this manuscript reaches the quality of publication of Nature Communications. Several detailed comments have been included below.

Our response: Many thanks for the comments. We agree that the manuscript didn’t clearly present the novelty of this work, but we would like to explain as follows:

(1) Ref 34 emphasized that high-aspect-ratio structures play an important role in sensitivity and limit of detection (for measuring tiny pressures). Although the pressure-sensitive microstructure from Ref 34 was obtained from high-cost photolithography (wet etched silicon wafer), the sensor only exhibited a nonlinear maximum sensing range of 15 kPa with a peak sensitivity of 11.8 kPa^{-1} (not over a linear range). In contrast, we develop a simple and effective method to design and fabricate microstructures in the EDL-based capacitive pressure sensor with tunable sensitivity and linearity. The demonstrated sensor can achieve high sensitivity (33.7 kPa^{-1}) in an ultrabroad linear sensing range (1700 kPa).

(2) We have included 28 recent papers published in the recent 5 years in Supplementary Table 2 to compare the overall performance (Fig.1b updated as well).

(3) We have updated the second and third paragraphs in the introduction to discuss the recent progress of iontronic pressure sensors.

We also appreciate the invaluable suggestions from the reviewer to significantly improve the quality of our work.

Our modification to the manuscript:

(1) We have updated Fig. 1b.

(2) We have added Supplementary Table 2 to support Fig. 1b.

Supplementary Table 2. Performance comparison of different capacitive pressure sensors.

	Number	Type	Structure fabrication method	Sensitivity (kPa ⁻¹)	Linear Sensing Rang (kPa)	Linear sensing factor (S_p)	Pressure resolution	Response/recovery time (ms)	LOD (Pa)	Ref.
1	1	EDL: mold-based	Photolithography Silicon wafer	1.3	3	3.9	0.02% (base pressure of 5 kPa)	15/15	0.2	¹
	2	EDL: mold-based	Photolithography Silicon wafer	7.49	6	44.94	NR	9/9	0.9	¹⁸
	3	EDL: mold-based	PTFE template	7.7	7.4	56.98	NR	NR	NR	¹⁹
	4	EDL: mold-based	3D print mold	49.1	4-485	2.37×10^4	NR	0.61/3.63	NR	²⁰
2	5	EDL: structure-transfer	Transfer sandpaper structure	3302	10	3.302×10^4	0.0056% (base pressure of 320 kPa)	9/18	0.08	³
	6	EDL: structure-transfer	Transfer sandpaper structure	5.5	30	165	NR	70.4/92.8	2	²¹

	7	EDL: structure-transfer	Transfer sandpaper structure	9.17	0.013-2063	1.89×10^4	NR	5/16	13	22
	8	EDL: structure-transfer	Transfer sandpaper structure	6.94	100	694	NR	48/NR	2.88	23
	9	EDL: structure-transfer	Transfer sandpaper structure	200	60	1.2×10^4	1.4% (base weight of 71 kg)	98/70	20	24
	10	EDL: structure-transfer	Directly using fabric materials	6.5	10	65	NR	30/30	7.5	25
	11	EDL: structure-transfer	Directly using fabric materials	0.24	70	16.8	NR	18	35	26
3	12	Conventional: mold-based	Photolithography Silicon wafer	2.51	10	25.1	NR	84/117	2	4
	13	Conventional: mold-based	Photolithography Silicon wafer	0.022	5	0.11	NR	NR	NR	27
	14	Conventional: mold-based	Photolithography Silicon wafer	0.43	1	0.43	NR	33/33	3.4	28
	15	Conventional: mold-based	Photolithography Silicon wafer	7.68	0.5	3.84	NR	30/28	1 mg	29

	16	Conventional: mold-based	Commercially anodic aluminum oxide template	0.35	2	0.7	NR	48/60	4	30
	17	Conventional: mold-based	Commercially anodic aluminum oxide template	6.583	0.1	0.6583	NR	48/36	3	31
	18	Conventional: mold-based	Polyurethane sponge skeletons	0.062	0.3	18.6	NR	45/83	3	32
	19	Conventional: mold-based	Nickel foam template	3.13	1	3.13	NR	94/	0.07	33
	20	Conventional: mold-based	Micro-engraving plastic template	0.065	1700	111	1% (base pressure of 1000 kPa)	100/100	/	5
4	21	Conventional: structure-transfer	Transfer the outline of polystyrene spheres	30.2	0.13	3.926	NR	25/50	0.7	34
	22	Conventional: structure-transfer	Transfer the outline of polystyrene spheres	6.61	0.11	0.73	NR	100/100	1	35

23	Conventional: structure-transfer	Transfer the outline of reed leaves	0.6	1	0.6	NR	180/120	4.5	36
24	Conventional: structure-transfer	Transfer the outline of bamboo leaves	2.08	1	2.08	NR	500/700	20	37
25	Conventional: structure-transfer	Transfer the outline of lotus leaves	1.2	2	2.4	NR	36/58	0.8	38
26	Conventional: structure-transfer	Transfer the surface structure of obscured glass	1.1	0.5	0.55	NR	1000/NR	1	39
27	Conventional: structure-transfer	Directly using dried flower petal as dielectric layer	1.54	1	1.54	NR	NR	0.6	40
28	Conventional: structure-transfer	Transfer the surface structure of paper	0.62	2	1.24	NR	NR	6	41
	Our work	Laser-induced gradient pyramids	33.7	1700	5.729×10^4	0.00725% (base pressure of 2000 kPa)	6/11	0.36	

Note: NR (not reported, $S_P = S \cdot \Delta P$ (S : sensitivity and ΔP : the corresponding linear sensing range))

(3) We have updated the second and third paragraphs of the introduction.

Ionic liquids (ILs) of atomically thin (~1 nm) between positive and negative charges in the electron double layer (EDL) at the electrode/dielectric interface can significantly increase the piezocapacitive effect³. The EDL effect can be affected by the concentration of the ILs^{42, 43} and the measurement frequency⁴⁴. Mixing ILs with high-polarity polymers (e.g., PVDF-HFP^{2, 22, 23}, TPU⁴⁵/PU⁴⁶, and PVA^{15, 20}) to form a dielectric layer helps enhance the sensitivity of the microstructured pressure sensors^{13, 26, 47}. Using electrode materials with ions intercalation-based pseudocapacitance such as TiC₂Tx^{15, 24} in iontronic sensors can also enhance the capacitance and then the sensitivity. However, such high sensitivity is often limited to a small linear sensing range, with the sensitivity declining gradually as the pressure loading further increases.

High linearity over a broad sensing range usually comes at a cost of reduced sensitivity (e.g., 9280 kPa⁻¹ within 20 kPa⁴⁶ vs. 0.065 kPa⁻¹ within 1700 kPa⁵). The development of capacitive pressure sensors with high sensitivity in a wide linear range remains elusive due to the trade-off between sensitivity and linear range^{2, 20, 22}. Efforts to address this challenge lead to the exploration of ingenious structure designs in iontronic sensors. Representative examples include laminating multilayer microstructured ionic layer (sensitivity of 9.17 kPa⁻¹ over 2000 kPa)²² and adding microstructures on the surface of microdomes (sensitivity of 49.1 kPa⁻¹ within 485 kPa)²⁰. Despite the rapid developments, fabrication of the critical microstructures still relies on complex fabrication processes such as photolithography or transferring of random microstructures. Therefore, there is an urgent demand for a facile, effective, and low-cost fabrication method to prepare the programmable microstructure designed by quantitative numerical analysis.

This work, explores the low-cost CO₂ laser with a Gaussian beam to fabricate programmable structures such as gradient pyramidal microstructures (GPM) for iontronic sensors. The profile and height of each pyramid in the gradient array can be individually adjusted and optimized to provide even deformation as the pressure increases. Combining the programmable microstructures with an ultrathin IL layer results in an iontronic sensor to break the trade-off between sensitivity and linear sensing range, exhibiting a high sensitivity (33.7 kPa⁻¹) and excellent linearity (0.99) over an ultra-broad linear sensing range up to 1700 kPa.

1. The authors claimed that the sensor presented in this paper possesses the advantages of low fabrication cost. However, according to Figure S1, the fabrication of the gradient-structured electrode is extremely complicated and time-costed, so the low fabrication cost cannot be the major advantage of this sensor.

Our response: We appreciate the reviewer's suggestion. As the microstructure plays a vital role in the performance of pressure sensors, we tried to focus on the fabrication of pressure-sensitive microstructures. Compared with the commonly used method of photolithography for fabricating the designed microstructures, our method based on the cheap PMMA board and the CO₂ laser system (common in the machine shop) presents a low-cost fabrication method (Supplementary Table 1).

(i) Photolithography process:

Silicon nitride deposition→mask preparation→silicon etching→remove remaining silicon oxide→O₂ plasma treatment→trichloro-(1H,1H,2H,2H-perfluorooctyl) silane deposition

(ii) Our process:

Import the designed pattern into the laser system and set parameters to the corresponding color area and then start the laser system (see the following laser working interface for the fabrication time: 3'53" for pattern 1 and 5'43" for pattern 2).

Supplementary Table 1. Cost comparison between the wet etched silicon wafer and laser-ablated PMMA for fabricating the microstructures

Method	Materials	Facilities	Environment requirement	Time cost
Wet etching	Silicon wafer (4.0") \$13.9/piece, Photoresist, Silicon nitride; Potassium hydroxide; Trichloro- (1H,1H,2H,2H- perfluorooctyl) silane; IPA; Acetone;	Clean room, Lithography equipment, Spin coater, wet bench, O ₂ plasma;	High temperature	> 5 h ¹⁷
Our method	PMMA board (4.0") \$0.5/piece	CO ₂ laser system	Ambient conditions	10 min

Note: The costs for materials such as silicon wafers and PMMA boards are based on catalog prices (e.g., Amazon).

Our modification to the manuscript: We have updated the manuscript to emphasize the fabrication cost of the microstructures in the abstract and the “Fabrication and working mechanism” Section.

(1) “Here we present a facile fabrication method to integrate an ultrathin ionic layer and low-cost programmable gradient microstructures.”

(2) The superior performance of the iontronic pressure sensor relies on a low-cost gradient pyramidal structure of laser-ablated polymethyl methacrylate (PMMA) that can be created by a multi-step process (Supplementary Table 1).

2. In Figure 1b, the authors announced that conventional EDL-based sensor is poor at linearity. However, many papers (Nature Communications, 2020, 11,209; Adv. Funct. Mater. 2019, 1807343) have proved that the iontronic pressure sensor can also present high linearity even better than conventional capacitive sensors.

Our response: Thanks for the reviewer’s question. We have added a linear sensing factor S_P ($S_P = S \cdot \Delta P$ with S as the sensitivity and ΔP as the corresponding linear sensing range)²⁰ to discuss both sensitivity and linearity in Fig. 1b.

The paper (Nature Communications, 2020, 11, 209) shows different sensitivities in three separate sensing ranges: 3302.9 kPa⁻¹ in 0 to 10 kPa, 671.7 kPa⁻¹ in 10 to 100 kPa, and 229.9 kPa⁻¹ in 100 to 360 kPa. Also, the microstructure transferred from the sandpaper is extremely irregular and random, which lacks repeatability and cannot be well controlled.

The other paper (Adv. Funct. Mater. 2019, 1807343) exhibits a sensitivity of 2.26 nF/kPa in the pressure range from 0 to 25 kPa. The linear sensing range

is very limited compared to conventional parallel^{6, 25, 48} or EDL-based capacitors^{2, 23, 26}. Although the paper was used to support the dielectric layer with a low concentration of ILs, the analysis assumed a uniform distribution of paper fibers and only considered the mechanical behavior.

Our modification to the manuscript: We have combined sensitivity and linear sensing range into the linear sensing factor in the updated Figure 1b.

3. Figure 1a is confusing, and we still don't understand the detailed structure of the sensor according to this image.

Our response: We appreciate the reviewer's careful check. We have updated Fig. 1a with the layer information.

Our modification to the manuscript: We have updated Fig. 1a.

4. Most importantly, a theoretical model should be studied to guide the structure design. Why the gradient microstructure could achieve a higher linearity? A theoretical equation is necessary. Current study only focus on the phenomenon.

Our response: We thank the reviewer for the suggestion. The normalized cross-section w of the contact surface of the pyramid microstructure increases proportionally to the square root of the compressive force F against the pyramid⁴⁹.

$$w \propto \sqrt{F}$$

(1)

The capacitance C is directly proportional to the area or square of the cross-section w of the contact surface. Therefore, the capacitance becomes directly proportional to the compressive force.

$$C \propto F \tag{2}$$

The linear dependence of capacitance and compressive force originates from the non-linear relationship between the cross-section and compressive force in Eq. (1). For the gradient microstructure, the effective cross-section w_{eff} of

the contact increases in a cascading order for each new pillar $i \leq N$ after exceeding every incremental force π_i . The incremental force π_i depends on the gradient of the microstructure: as the gradient increases, more exceeding force π_i is required to start the deformation of the i^{th} pillar. For example, the incremental force will be zero for all the pillars in a uniform pillar distribution. In comparison, for a gradient pillar distribution, only the exceeding force π_1 corresponding to the initially deformed pillar $i = 1$ will be zero, and this force increases with the pillar index $\pi_i < \pi_{i+1}$ ($\forall i \leq N$). The force F_i deforms each pillar i with width w_i according to the following relationship:

$$w_i(F) = \begin{cases} w_i \propto \sqrt{F_i}, & F > \pi_i \\ 0, & F \leq \pi_i \end{cases}$$

Therefore, the effective cross-section w_{eff} is given by

$$w_{\text{eff}} = \sum_{i=1}^{N(F)} w_i(F) \propto \sum_{i=1}^{N(F)} \sqrt{F_i} \geq \sqrt{\sum_{i=1}^{N(F)} F_i} \approx \sqrt{N(F)\langle F \rangle}, \quad (3)$$

where $N(F)$ is the number of pillars deformed with $\pi_N < F < \pi_{N+1}$ and average force $\frac{1}{N(F)} \sum_{i=1}^{N(F)} F_i = \langle F \rangle$.

The effective capacitance C_{eff} is thus given by

$$\frac{\partial C_{\text{eff}}}{\partial \langle F \rangle} \geq kN(F), \quad (4)$$

where k is the proportional constant. As the effective capacitance depends on the square of the effective cross-section, the slope of the capacitance force is lower-bounded by the number of pillars N . The gradient microstructure increases the exceeding force π_i to delay the deformation of the pillars due to the compressive force, resulting in a more linear range.

Our modification to the manuscript: We have added the theoretical analysis in the revised Supplementary Information.

Moreover, the GPML exhibits higher sensitivity than that of UPM and GPMS (Fig. 3d and Supplementary Note 2).

5. The pyramidal structures are too rough, a finer method should be used to fabricate the structure.

Our response: Thanks for the reviewer's question and suggestion. The pyramid-like structure created in this work can be regarded as the stacking of multiple cubes with gradually decreased dimensions. The tri-layered PMMA template was fabricated by decreasing laser power accompanied by simultaneously increased square dimension in each successive layer. Despite the rough surface created by the low-cost CO₂ laser with a low resolution, the pyramid-like microstructure can still give a small tip area for high sensitivity and evenly deform under pressure for a large sensing range. In fact, the resulting sensor outperforms those based on the pyramidal microstructures obtained from the silicon wafer template (Supplementary Table 2). It is possible that the finer microstructures may not necessarily correlate to the improved pressure sensor performance. It is also worth noting that the rough surface (e.g., structures taken from sandpapers²³, leaf³⁸, and obscured glass³⁹) has been leveraged to enhance the performance²⁰. Meanwhile, the fine features from the rough surface in our pyramid-like microstructure also contribute to the ultralow detection limit of 0.36 Pa.

Our modification to the manuscript: We have added the discussion on the possibility to achieve a pyramid-like structure with more accurate features in the fabrication sections.

To achieve this, decreased laser power is accompanied by simultaneously increased square laser pattern in each successive layer for fabricating the tri-layered PMMA template and corresponding PMDS pyramidal microstructure (Supplementary Table 3 d-f and Supplementary Fig. 4a-c). The pyramid-like microstructure with more accurate features is possible to achieve with an increased number of laser ablation layers.

6. The sensitivity of the sensor is highly related with the initial capacitance of the sensor. The initial capacitances of the sensors in Figure 2d and 2e should be informed. If the high sensitivity of the sensor derived from the extremely low initial capacitance? If yes, why the sensor has such low initial capacitance?

Our response: We appreciate the reviewer's suggestion and question. The high sensitivity of the sensor did derive from the low initial capacitance. We had explained that in the "Results" section of the manuscript.

Fabrication and working mechanism

"Meanwhile, the gradient pyramid also minimizes the initial contact area (before pressure) at the structured electrode/IL interface to result in a larger normalized contact area and capacitance change upon pressure loading".

Sensing performance modulated by the enhanced EDL effect

"The initially small contact area at the fine tip of the pyramid created from the laser ablated rough surface (Supplementary Fig. 10a) further provides the iontronic pressure sensor with a low limit detection of 0.36 Pa to detect a light tissue (Fig. 5c)."

Our modification to the manuscript: We have added the initial capacitance

of the sensor with different microstructures.

Therefore, the resulting sensor exhibits a much higher response than that with uniform pyramidal microstructures (UPM) or other microstructures (Fig. 2f), due to the reduced initial capacitance from 823 (square frustum) to 43 (UPM) and then to 12 pF (gradient pyramidal microstructures with large height differences (GPML)).

7. In figure 3, how to determine the initial contact area A_0 of different structures?

Our response: We appreciate the reviewer's question. The initial contact area A_0 is the area of the microstructure tip, which was estimated from the side view of SEM images (e.g., Fig. 2b).

Our modification to the manuscript: We have added a new sentence in the "Methods-Finite element analysis (FEA)" Section to explain the estimation of the initial contact area.

The pressure of 1500 kPa was applied on the top electrode that was treated as a rigid plate (not shown in Fig. 3a). The initial contact area was estimated from SEM images.

8. In figure 5b, finger touch is not a quantitative method to evaluate the response time. Please using the standard method described in other papers.

Our response: We appreciate the reviewer's suggestions. We have updated Fig. 5b using the standard method (i.e., using a weight of 50 g).

Our modification to the manuscript: Besides the high sensitivity over the ultrabroad pressure range, the sensor with the gradient microstructures also exhibits a fast response/recovery of 6/11 ms from a pressure loading of 5 kPa (Fig. 5b).

9. In figure 5c, we still cannot judge if the fluctuation of the curve comes from the noise or the tissue.

Our response: We thank the reviewer for the careful check. We have retested the detection limit and updated Fig. 5c (with an improved signal-to-noise ratio).

Our modification to the manuscript: We have updated Fig. 5c.

10. *If the high linearity of the sensor can be retained when bending? Seems that the high linearity cannot be achieved when the pressure is applied non-uniformly.*

Our response: We appreciate the reviewer’s question. We agree with the reviewer that bending results in uneven pressure loading (further leading to the buckling of the pyramid). We have added a new paragraph in Section “Sensing performance modulated by the enhanced EDL effect” and a corresponding plot (Supplementary Fig. 9b) to discuss the sensitivity of the sensor under different bending conditions.

Our modification to the manuscript: We have added a new paragraph in the “Sensing performance modulated by the enhanced EDL effect” Section and a corresponding plot (Supplementary Fig. 9b).

The flexibility of the sensor allows it to easily attach to a curved surface (Supplementary Figure 9a). The outstanding performance of the pressure sensor (e.g., high sensitivity and linearity) usually benefits from ingenious

structure designs. However, the sensor performance is often significantly reduced under bending conditions¹⁵ likely due to the change in the microstructures. As the bending radius is decreased from 45 to 31 and then to 13 mm, the sensitivity of GPML₇₀₀ decreases from 11.96 to 6.10 and then to 1.86 kPa⁻¹ (Supplementary Figure 9b). This decrease results from the deformation in the highest pyramid and the reduced gap between the structured bottom electrode and the dielectric layer, increasing the initial capacitance (before pressure loading). The increased initial capacitance reduces the capacitance changes of the sensor under pressure and thus the sensitivity. Additionally, the bending strain also causes an uneven pressure loading on lower pyramids due to deformed pyramids, greatly reducing the linear sensing range to ca. 500 kPa compared with 1700 kPa before bending. However, the iontronic pressure sensor still exhibits reasonably good sensitivity in a relatively large linear sensing range. Further improvement can be possible when the iontronic pressure sensor is directly fabricated on the 3D freeform surfaces with a modified laser setup¹⁶.

Supplementary Figure 9. b) The pressure sensing performance of the iontronic pressure sensor under different bending conditions.

Reviewer #4:

Manuscript titled " Iontronic pressure sensor with ultra-broad linear range and high sensitivity enabled by laser-induced gradient pyramidal structures" is a nice work related to Development and testing of system for pressure sensing application.

Recommendation: Review Again After Resubmission (Paper is not acceptable in its current form, but has merit. A major rewrite is required. Author should be encouraged to resubmit a rewritten version after the changes suggested in the Comments section have been completed.) Before publishing, the authors must revise the paper as per comments mentioned below:

Our response: We thank the reviewer for the highly positive evaluation of our work. We also appreciate the insightful comments that help significantly improve the overall quality of this work.

1. Authors have developed and tested a system for pressure sensing application which is of good quality. The work supports the conclusions and claims. Data analysis, interpretation and conclusions presentation is satisfactory. Methodology is fine.

Our response: We thank the reviewer for the highly positive evaluation of our work.

Our modification to the manuscript: None

2. The work is significant and can be considered for publication in the journal after some improvements as suggested next.

Our response: We appreciate the reviewer's suggestion. We have addressed

the comments in the following, with all revisions highlighted in yellow in the revised manuscript.

Our modification to the manuscript: We have addressed the comments in the following, with all revisions highlighted in yellow in the revised manuscript.

3. Is there enough detail provided in the methods for the work to be reproduced? Sensing range of 1700 kPa is quite good. What about its Low and High Limits? Can the low limit be decreased further in future?

Our response: Many thanks for the reviewer's questions. We have provided additional details in the updated the "**Methods-Preparation of top electrode and iontronic layer**" Section and in Supplementary Table 3.

The low limit is 0.36 Pa (Fig. 5c) and there are no changes in the capacitance as the pressure exceeds 1700 kPa for the pressure sensor with the IL concentration of 50 wt%. However, modulation in the concentration of IL can be used to increase the high limit. For instance, using a lower IL concentration of 20 wt% gives a broader sensing range of up to 3000 kPa (Fig. 4ef).

As for the low limit, it is possible to introduce an air gap between the electrode and dielectric layer to further reduce the low limit²⁴.

Our modification to the manuscript:

(1) We have added the discussion of possible methods to further modulate the sensor performance in the “**Discussion**” Section.

The represented demonstrations presented in this study open up opportunities for future wearable health-monitoring devices, smart prosthesis skins, and intelligent human-machine collaboration systems.

Despite the high sensitivity in a broad linear sensing range, further performance improvements are possible. For example, the sensitivity can be enhanced by increasing the concentration of ILs and the sensing range can be extended by using larger pyramids for more significant contact area changes. The detection limit can be further decreased by including an air gap between the electrode and the dielectric layer. Further optimization of the laser parameters to create various height combinations in the microstructures may help modulate the sensor performance, which will be pursued in our future studies.

(2) We have added additional details in the “**Methods**” Section.

Preparations of the top electrode and iontronic layer: After preparing the acetone and PVDF-HFP with a mass ratio of 20:3, the IL [EMI][TFSI] was added with a mass ratio of 20%, 35%, or 50% of PVDF-HFP to obtain the mixture. The mixture was heated and stirred at 120 °C and 1000 rpm until PVDF-HFP was completely dissolved to obtain the uniform solution. The solution was then spin-coated on the as-received ITO/PET film (cut to 1.2 cm×1.2 cm) at 1000 rpm for 30 s and dried on the hot plate at 80°C in a fume hood to fully evaporate the acetone and cure the iontronic layer.

(3) We have updated Supplementary Table 3.

Supplementary Table 3. Different designs in the laser ablation pattern for varying microstructures

Number	Pattern	Laser power	Side length/diameter (μm)	Distribution
a (S2-b)		30%	500	b (S2-c)		30%	250	c (S2-d)		30%	700	d (S4-d)		30%/25%/20% from inside to outside	704/563/422	e (S4-e)		30%/25%/20% from inside to outside	526/421/316	f (S4-f)		30%/25%/20% from inside to outside	352/282/211	GPMS ₅₀₀		25%/21%/14% black/red/purple		30%/25%/20% yellow/blue/pink
GPML ₅₀₀		25%/10%/10% black/red/purple		30%/25%/20% yellow/blue/pink
GPMS ₇₀₀		25%/21%/14% black/red/purple		30%/25%/20% yellow/blue/pink
GPML ₇₀₀		25%/10%/10% black/red/purple		30%/25%/20% yellow/blue/pink

4. “The development of capacitive pressure sensors with high sensitivity in a wide linear range remains elusive due to the trade-off between sensitivity and linear range.” Add some quantitative data from published literature here to support this statement. Last paragraph of Introduction Section should highlight research gaps, and main objectives of this work.

Our response: We thank the reviewer for the suggestion. We have included the recent developments of the iontronic pressure sensor and discussed the research gaps and main objectives in the updated introduction.

Our modification to the manuscript: We have updated the introduction section as follows.

Ionic liquids (ILs) of atomically thin (~1 nm) between positive and negative charges in the electron double layer (EDL) at the electrode/dielectric interface can significantly increase the piezo-capacitive effect³. The EDL effect can be affected by the concentration of the ILs^{42, 43} and the measurement frequency⁴⁴. Mixing ILs with high-polarity polymers (e.g., PVDF-HFP^{2, 22, 23}, TPU⁴⁵/PU⁴⁶, and PVA^{15, 20}) to form a dielectric layer helps enhance the sensitivity of the microstructured pressure sensors^{13, 26, 47}. Using electrode materials with ions intercalation-based pseudocapacitance such as TiC₂Tx^{15, 24} in iontronic sensors can also enhance the capacitance and then the sensitivity. However, such high sensitivity is often limited to a small linear sensing range, with the sensitivity declining gradually as the pressure loading further increases.

High linearity over a broad sensing range usually comes at a cost of reduced sensitivity (e.g., 9280 kPa⁻¹ within 20 kPa⁴⁶ vs. 0.065 kPa⁻¹ within 1700 kPa⁵). The development of capacitive pressure sensors with high sensitivity in a wide linear range remains elusive due to the trade-off between sensitivity and linear range^{2, 20, 22}. Efforts to address this challenge lead to the exploration of ingenious structure designs in iontronic sensors. Representative examples include laminating multilayer microstructured ionic layer (sensitivity of 9.17 kPa⁻¹ over 2000 kPa)²² and adding microstructures on

the surface of microdomes (sensitivity of 49.1 kPa^{-1} within 485 kPa)²⁰. Despite the rapid developments, fabrication of the critical microstructures still relies on complex fabrication processes such as photolithography or transferring of random microstructures. Therefore, there is an urgent demand for a facile, effective, and low-cost fabrication method to prepare the programmable microstructure designed by quantitative numerical analysis.

This work, explores the low-cost CO_2 laser with a Gaussian beam to fabricate programmable structures such as gradient pyramidal microstructures (GPM) for iontronic sensors. The profile and height of each pyramid in the gradient array can be individually adjusted and optimized to provide even deformation as the pressure increases. Combining the programmable microstructures with an ultrathin IL layer results in an iontronic sensor to break the trade-off between sensitivity and linear sensing range, exhibiting a high sensitivity (33.7 kPa^{-1}) and excellent linearity (0.99) over an ultra-broad linear sensing range up to 1700 kPa .

5. Restructuring of the paper is required for easy readability

Our response: We appreciate the reviewer's suggestion.

Our modification to the manuscript: We have restructured the “Results” section.

6. Fig. 1 shows the methodology. Authors are advised to Bifurcate Section 2 into Two Major Sections, that is, Section 2: Materials and Methods. Section 3: Results and Discussion. Fig. 1 can then be good for Section 2: Materials and Methods

Our response: We appreciate the reviewer's suggestions. We have carefully checked the guideline of Nature Communications and kept the Methods section

at the end of this manuscript per guideline requirements.

Our modification to the manuscript: We have added a new title for Fig.1 and Fig. 2. We have also renumbered the subtitles of the “**Results**” Section.

Results

Fabrication and working mechanism

Gradient pyramidal microstructures from laser-ablated molds

Sensing performance of different laser-induced structures

Sensing performance modulated by the enhanced EDL effect

Detections of static and dynamic subtle motions

High pressure resolution under large pressure preloads

Interactive robotic hand with closed-loop control

7. *FEA is good.*

Our response: We thank the reviewer for the positive evaluation of our simulation efforts.

Our modification to the manuscript: None

8. *Figure 3a: Mpa should be MPa*

Our response: We thank the reviewer for the careful check and we have corrected the typo in Fig. 3a.

Our modification to the manuscript: We have replaced Mpa with MPa.

9. Section 4.4: the authors should mention that how much displacement is applied on the top electrode? Is it a pressure load? If it is so, please mention the magnitude.

Our response: We thank the reviewer for the question. We have corrected the typo and updated it to “a pressure of 1500 kPa”.

Our modification to the manuscript: We have corrected the typo.

“A pressure of 1500 kPa was applied on the top electrode that was treated as a rigid plate (not shown in Fig.3a).”

10. The authors should compare the sensitivities of different laser-induced structures used for the simulations, rather than just mentioning it for GPML only.

Our response: We appreciate the reviewer’s suggestions. We have compared the potential distribution and sensitivity of the uniform laser-induced structure and the gradient laser-induced structure. The uniform laser-induced structure showed a large initial capacitance, reducing the sensitivity by more than 220 times compared to that of the gradient structure.

Our modification to the manuscript: We have added this comparison in the revised manuscript and Supplementary Information.

“As a result, the increased sensitivity does not come at a cost of a significantly reduced linear sensing range when compared with the uniform structure (Supplementary Fig. 7).”

Supplementary Fig. 7. a Comparison in the potential distribution between the uniform (top) and gradient (bottom) structures. **b** The normalized relative capacitance changes as a function of the applied pressure between the gradient and uniform structures, with a 220-fold increase in the sensitivity for the gradient structure.

11. “The electric field distribution was computed by coupling mechanical, electrostatics,

and transport of dilute species modules.” Authors may add the governing mathematical model behind this coupling process and then it can be easily verified using some other standard software also such as MATLAB.

Our response: We appreciate the reviewer’s suggestions. The governing equations for electrostatics in the ionic liquid domain are given as follows:

$$\nabla \cdot D = \rho_v \quad (5)$$

$$D = \varepsilon_0 \det(F) (F^T F)^{-1} E + \varepsilon_0 \chi_r E + P(\varepsilon_{elastic}) \quad (6)$$

$$E = -\nabla V \quad (7)$$

where ρ_v is the formed free electron surface charge density and D is the electric displacement vector. The domain has a deformation gradient F , electric susceptibility of χ_r with a polarization of P in the elastic limit. The domain is under an electric field caused by the potential difference V between the boundaries of the domain. The governing equations for compressible neo-Hookean materials in the mechanical domain are given as follows:

$$0 = \nabla \cdot (FS)^T + F_v \quad (8)$$

$$S = \frac{\partial W_s}{\partial \varepsilon} \quad (9)$$

$$\varepsilon = \frac{1}{2} (F^T F - I) \quad (10)$$

$$W_s = \frac{1}{2} \mu (I_1 - 3) - \mu \log(J_{elastic}) + \frac{1}{2} \lambda \log(J_{elastic})^2 \quad (11)$$

where W_s is the elastic strain energy density that is a function of elastic strain state ε . For a compressible neo-Hookean material, the elastic strain energy depends on the elastic volume ratio $J_{elastic}$, Lamé parameters μ, λ , and the first invariant of the elastic right Cauchy-Green deformation tensor I_1 . The governing equations for the transport of dilute species in the ionic liquid domain are given as follows:

$$\nabla \cdot J = R \quad (12)$$

$$J = -D\nabla c - zu_m F_a c \nabla V \quad (13)$$

where the concentration c of each species and diffusion constant D contribute to the current J . The other contribution comes from the migration of species of charge z and mobility u_m due to the electric potential V . The space charge coupling between electrostatics and the transport of dilute species is given by the following governing equations:

$$\nabla \cdot D = \rho_v \quad (14)$$

$$\rho_v = F_a \sum_i z_i c_i \quad (15)$$

where F_a is Faraday's constant.

Our modification to the manuscript: We have added governing equations of the coupling process in Supplementary Note 3. It is difficult to verify the coupling process with standard software such as MATLAB due to the complex geometry in the pyramidal microstructures. Therefore, the coupling process was solved numerically with the relevant commercial software (i.e., COMSOL Multiphysics 5.6) in this work.

“The electric field distribution was computed by coupling mechanical, electrostatics, and transport of dilute species modules (Supplementary Note 3).”

12. “Increasing the pressure and concentration at ITO electrode resulted in the increased electric field distribution, which further facilitated the transport of the ions in the iontronic medium.” Does increase in pressure only facilitates or increases the rate of transport of ions also?

Our response: We appreciate the reviewer's question. With the increase in

the pressure, the polarization P in the ionotropic medium increases, which contributes to the electric displacement vector D in Eq. (6). The divergence of electric displacement vector D increases the free electron surface charge density in Eq. (5). The free electron surface charge density in turn contributes to the migration of species as shown in the space-charge coupling equation (15). Thus, the increases in pressure will facilitate the transport of ions.

It is not obvious that the rate of transport of ions will be increasing with the application of steady pressure. However, time-varying pressure may contribute to a time-varying electric displacement field, which may result in time-varying free surface charge density and thus result in an increased rate of transport of ions.

Our modification to the manuscript: We have changed “facilitated” into “generated”. We have also discussed the potential change in the rate of transport of ions.

“The transport of dilute species was **generated** by the created electric field from the applied potential on the top electrode and the created polarization field due to the pressure applied on piezoelectric PVDF”. On the other hand, the transport of the diluted species generated an electric field due to the space charge density coupling. Increasing the pressure and concentration at ITO electrode resulted in the increased electric field distribution, which further facilitated the transport of the ions in the ionotropic medium. **It is worth noting that time-varying pressure may contribute to a time-varying electric displacement field, which may result in time-varying free surface charge density for an increased rate of transport of ions.**

References:

1. Zhu P, *et al.* Skin-electrode iontronic interface for mechanosensing. *Nat Commun* **12**, 4731 (2021).
2. Lu P, *et al.* Iontronic pressure sensor with high sensitivity and linear response over a wide pressure range based on soft micropillared electrodes. *Science Bulletin* **66**, 1091-1100 (2021).
3. Bai N, *et al.* Graded intrafillable architecture-based iontronic pressure sensor with ultra-broad-range high sensitivity. *Nat Commun* **11**, 209 (2020).
4. Zhang Z, *et al.* Highly sensitive capacitive pressure sensor based on a micropylramid array for health and motion monitoring. *Advanced Electronic Materials* **7**, 2100174 (2021).
5. Ji B, *et al.* Gradient Architecture-Enabled Capacitive Tactile Sensor with High Sensitivity and Ultrabroad Linearity Range. *Small* **17**, e2103312 (2021).
6. Ji B, Zhou Q, Hu B, Zhong J, Zhou J, Zhou B. Bio - inspired hybrid dielectric for capacitive and triboelectric tactile sensors with high sensitivity and ultrawide linearity range. *Adv Mater* **33**, 2100859 (2021).
7. Deng W, *et al.* Microstructure-based interfacial tuning mechanism of capacitive pressure sensors for electronic skin. *Journal of Sensors* **2016**, (2016).
8. Tee BCK, Chortos A, Dunn RR, Schwartz G, Eason E, Bao Z. Tunable flexible pressure sensors using microstructured elastomer geometries for intuitive electronics. *Adv Funct Mater* **24**, 5427-5434 (2014).
9. He B, Yan Z, Zhou Y, Zhou J, Wang Q, Wang Z. FEM and experimental studies of flexible pressure sensors with micro-structured dielectric layers. *J Micromech Microeng* **28**, 105001 (2018).

10. Zhu B, *et al.* Hierarchically Structured Vertical Gold Nanowire Array-Based Wearable Pressure Sensors for Wireless Health Monitoring. *ACS Appl Mater Interfaces* **11**, 29014-29021 (2019).
11. Yang JC, *et al.* Microstructured Porous Pyramid-Based Ultrahigh Sensitive Pressure Sensor Insensitive to Strain and Temperature. *ACS Appl Mater Interfaces* **11**, 19472-19480 (2019).
12. Mannsfeld SC, *et al.* Highly sensitive flexible pressure sensors with microstructured rubber dielectric layers. *Nature materials* **9**, 859-864 (2010).
13. Cho SH, *et al.* Micropatterned pyramidal ionic gels for sensing broad-range pressures with high sensitivity. *ACS applied materials & interfaces* **9**, 10128-10135 (2017).
14. Qiu Y, *et al.* Bioinspired, multifunctional dual-mode pressure sensors as electronic skin for decoding complex loading processes and human motions. *Nano Energy* **78**, 105337 (2020).
15. Luo Y, *et al.* Gecko-inspired slant hierarchical microstructure-based ultrasensitive iontronic pressure sensor for Intelligent interaction. *Research* **2022**, (2022).
16. Zheng B, Zhao G, Yan Z, Xie Y, Lin J. Direct Freeform Laser Fabrication of 3D Conformable Electronics. *Adv Funct Mater*, 2210084 (2022).
17. Lee S, *et al.* A highly sensitive bending sensor based on controlled crack formation integrated with an energy harvesting pyramid layer. *Advanced Materials Technologies* **3**, 1800307 (2018).
18. Lu P, *et al.* Iontronic pressure sensor with high sensitivity and linear response over a wide pressure range based on soft micropillared electrodes. *Science Bulletin* **66**, 1091-1100 (2021).

19. Zhou H, *et al.* Capacitive pressure sensors containing reliefs on solution-processable hydrogel electrodes. *ACS Applied Materials & Interfaces* **13**, 1441-1451 (2021).
20. Bai N, *et al.* Graded Interlocks for Iontronic Pressure Sensors with High Sensitivity and High Linearity over a Broad Range. *ACS nano* **16**, 4338-4347 (2022).
21. Sharma S, *et al.* Hydrogen-Bond-Triggered Hybrid Nanofibrous Membrane-Based Wearable Pressure Sensor with Ultrahigh Sensitivity over a Broad Pressure Range. *ACS nano* **15**, 4380-4393 (2021).
22. Xiao Y, *et al.* Multilayer double-sided microstructured flexible iontronic pressure sensor with a record-wide linear working range. *ACS sensors* **6**, 1785-1795 (2021).
23. Zheng Y, *et al.* Highly sensitive electronic skin with a linear response based on the strategy of controlling the contact area. *Nano Energy* **85**, 106013 (2021).
24. Gao L, *et al.* Highly sensitive pseudocapacitive iontronic pressure sensor with broad sensing range. *Nano-micro letters* **13**, 1-14 (2021).
25. Lin Q, *et al.* Highly sensitive flexible iontronic pressure sensor for fingertip pulse monitoring. *Advanced Healthcare Materials* **9**, 2001023 (2020).
26. Shen Z, Zhu X, Majidi C, Gu G. Cutaneous ionogel mechanoreceptors for soft machines, physiological sensing, and amputee prostheses. *Adv Mater* **33**, 2102069 (2021).
27. Ruth SRA, Beker L, Tran H, Feig VR, Matsuhisa N, Bao Z. Rational design of capacitive pressure sensors based on pyramidal microstructures for specialized monitoring of biosignals. *Adv Funct Mater* **30**, 1903100 (2020).

28. Luo Z, *et al.* High-resolution and high-sensitivity flexible capacitive pressure sensors enhanced by a transferable electrode array and a micropillar–PVDF film. *ACS Applied Materials & Interfaces* **13**, 7635-7649 (2021).
29. Yang J, *et al.* Flexible, Tunable, and Ultrasensitive Capacitive Pressure Sensor with Microconformal Graphene Electrodes. *Acs Applied Materials & Interfaces* **11**, 14997-15006 (2019).
30. Guo Y, Gao S, Yue W, Zhang C, Li Y. Anodized aluminum oxide-assisted low-cost flexible capacitive pressure sensors based on double-sided nanopillars by a facile fabrication method. *ACS applied materials & interfaces* **11**, 48594-48603 (2019).
31. Niu H, Gao S, Yue W, Li Y, Zhou W, Liu H. Highly morphology - controllable and highly sensitive capacitive tactile sensor based on epidermis - dermis - inspired interlocked asymmetric - nanocone arrays for detection of tiny pressure. *Small* **16**, 1904774 (2020).
32. Qiu J, *et al.* Rapid-response, low detection limit, and high-sensitivity capacitive flexible tactile sensor based on three-dimensional porous dielectric layer for wearable electronic skin. *ACS applied materials & interfaces* **11**, 40716-40725 (2019).
33. Ha KH, *et al.* Highly Sensitive Capacitive Pressure Sensors over a Wide Pressure Range Enabled by the Hybrid Responses of a Highly Porous Nanocomposite. *Adv Mater* **33**, e2103320 (2021).
34. Xiong Y, *et al.* A flexible, ultra-highly sensitive and stable capacitive pressure sensor with convex microarrays for motion and health monitoring. *Nano Energy* **70**, (2020).
35. Zhang Y, *et al.* Flexible and Highly Sensitive Pressure Sensors with Surface Discrete Microdomes Made from Self - Assembled Polymer Microspheres Array. *Macromol Chem Phys* **221**, 2000073 (2020).

36. Liu Y-Q, Zhang J-R, Han D-D, Zhang Y-L, Sun H-B. Versatile electronic skins with biomimetic micronanostructures fabricated using natural reed leaves as templates. *ACS applied materials & interfaces* **11**, 38084-38091 (2019).
37. Liu Z, *et al.* Natural bamboo leaves as dielectric layers for flexible capacitive pressure sensors with adjustable sensitivity and a broad detection range. *RSC Advances* **11**, 17291-17300 (2021).
38. Wan Y, *et al.* A highly sensitive flexible capacitive tactile sensor with sparse and high - aspect - ratio microstructures. *Advanced Electronic Materials* **4**, 1700586 (2018).
39. Quan Y, *et al.* Highly sensitive and stable flexible pressure sensors with micro-structured electrodes. *J Alloy Compd* **699**, 824-831 (2017).
40. Wan Y, *et al.* Natural plant materials as dielectric layer for highly sensitive flexible electronic skin. *Small* **14**, 1801657 (2018).
41. Lee K, *et al.* Rough - surface - enabled capacitive pressure sensors with 3D touch capability. *Small* **13**, 1700368 (2017).
42. Zheng Y, *et al.* Highly sensitive electronic skin with a linear response based on the strategy of controlling the contact area. *Nano Energy* **85**, (2021).
43. Li S, *et al.* All - in - One Iontronic Sensing Paper. *Adv Funct Mater* **29**, 1807343 (2019).
44. Insang You DGM, Naoji Matsuhisa, Jiheong Kang, Jimin Kwon, Levent Beker, Jaewan Mun, Wonjeong Suh, Tae Yeong Kim, Jeffrey B.-H. Tok, Zhenan Bao, Unyong Jeong. Artificial multimodal receptors based on ion relaxation dynamics. (2020).
45. Cui X, *et al.* Flexible and breathable all-nanofiber iontronic pressure sensors with ultraviolet shielding and antibacterial performances for wearable electronics. *Nano Energy* **95**, 107022 (2022).

46. Liu Q, Liu Y, Shi J, Liu Z, Wang Q, Guo CF. High-Porosity Foam-Based Iontronic Pressure Sensor with Superhigh Sensitivity of 9280 kPa⁻¹. *Nano-Micro Letters* **14**, 1-12 (2022).
47. Chhetry A, Kim J, Yoon H, Park JY. Ultrasensitive Interfacial Capacitive Pressure Sensor Based on a Randomly Distributed Microstructured Iontronic Film for Wearable Applications. *ACS Appl Mater Interfaces* **11**, 3438-3449 (2019).
48. Jung Y, *et al.* Linearly sensitive pressure sensor based on a porous multistacked composite structure with controlled mechanical and electrical properties. *ACS Applied Materials & Interfaces* **13**, 28975-28984 (2021).
49. Liu S, Olvera de la Cruz M. Deformation of elastomeric pyramid pen arrays in cantilever - free scanning probe lithography. *Journal of Polymer Science Part B: Polymer Physics* **56**, 731-738 (2018).

REVIEWER COMMENTS

Reviewer #1 (Remarks to the Author):

The issues were well addressed. The manuscript can be published in NC.

Reviewer #2 (Remarks to the Author):

The revised version of manuscript NCOMMS-22-32774A entitled "Iontronic pressure sensor with high sensitivity over ultra-broad linear range enabled by laser-induced gradient micro-pyramids" by Yang et al., addresses most of my previous concerns and is substantially improved over the previous version.

I still disagree in the use of low-cost to qualify these sensors. The authors try to justify claims of low cost by introducing Supplementary Table 1, which compares wet-etched silicon wafer and laser ablated PMMA. While I appreciate the author's efforts, the comparison is mostly qualitative and not supported by references or other objective data. A true estimation of cost will require cost of ownership of equipment, operation, depreciation, in addition to throughput and yield calculations, noting that lithography is a high throughput parallel process while laser etching is based on a low throughput raster process. I do not think that claiming "low-cost" (I would frame it as: "reduced cost compared with...") is necessarily wrong, but is clearly not supported by objective data and a clear methodology to estimate it. In my view, is also not necessary given the advances reported in this paper.

A few other minor deficiencies need to be corrected.

On the Introduction, page 2, revise grammar following text: "Ionic liquids (IL) of atomically thin (~ 1 nm) between positive and negative charges...." Perhaps "Atomically thin ionic liquid layers (~ 1 nm)...."

On page 3, replace the word "ingenious" with "various" or "novel" in the phrase: "Efforts to address this challenge lead to the exploration of ingenious structure designs in iontronic sensors."

On page 9, replace "Supplementary Fig. 4de" with: "Supplementary Fig. 4d-e"

Reviewer #3 (Remarks to the Author):

Although the revised manuscript has responded to many critical comments in the revised form, the fundamental issues presented in the current form are still not fully addressed. Particularly, the mechanism to achieve outstanding performance (e.g., high linearity and high performance) is not well explained; the low-cost processing remains unclear or not convinced; the high sensitivity and SNR results are lack of critical information; the fabrication process with laser-engraved microstructures may be very difficult to be repeated with high consistence to satisfy real high-performance medical needs; and at last, the applications proposed have been widely investigated in multiple prior arts. Therefore, it still has not yet reached the quality of innovation and performance criteria of Nature Communications in its revised form. The detailed comments have been included below:

1. The main concept of height-gradient structure to increase the device linearity presented is quite similar to ref. 9 in this manuscript, which uses gradient micro-dome array to achieve high linearity in a capacitive counterpart. Unfortunately, this paper does not offer a comprehensive comparison on its performance with that of ref.9; in particular, the shape influence of the microstructure vs the device performance. Moreover, the theory behind its mechanism presented in the rebuttal version is not highly relevant to its performance. Specifically, in A4 of the rebuttal letter, the claimed linear relationship between capacitive output of the sensor and the applied pressure cannot be derived from Eq. 3 and Eq. 4. Moreover, it is confusing that the proposed equations in the revision do not explain/predict the performance, including linearity range, sensitivity, etc.

2. Secondly, the claimed low-cost feature is not very solid in the practical view. Particularly, in A1 of the rebuttal letter, the authors have only compared the cost of the sensors with that prepared

by lithography process and acknowledged as a high-cost approach. The comparisons with the other low-cost processes, for instance, molding, printing, etc. have been largely ignored. However, the proposed sensor structure in this manuscript involves a sputtered gold layer as the electrode, which is normally considered as an expensive step, in comparison with the screen-printing or electroplated process with a much lower cost of metal compounds.

3. It is highly skeptical on the high-performance demonstration of the sensor in terms of the resolution, in which a value of 0.00725% under ultrahigh pressure of 2000 kPa was announced. However, the ballistocardiograph signals of a typically human subject is not observed (similar to a pulse wave signal), which is quite surprising, because the magnitude of the signals should be considerably greater than that of the sensor resolution claimed in the manuscript.

4. In A8, the author shows interesting measurement results to load and unload an object with 50g in several milliseconds. It is extremely difficult to conduct such an experiment to the reviewer's knowledge. A detailed description of the test, including a video clip, should be presented to demonstrate such a weight-loading experiment.

5. In A9, the results presented in the revised version shows a significant improvement on the device SNR from its original article, however, it is not well explained how the conducted experiment differs from the original (in Fig. 5c), why does it now show significantly better outcomes in SNR in the revision.

6. In the application of "High pressure resolution under larger pressure preloads", the calculated pressure change of dropping pens is incorrect. The result based on its data should yield 145Pa, but not 14.5Pa, leading to the flawed conclusion that "The extraordinarily high pressure resolution of 0.00725% under a huge pressure load of 2 MPa significantly outperforms the commercial weight scale (Supplementary Fig. 10c), human skin (~ 7%), and other microstructured pressure sensors previously reported in the literature". If corrected, the reviewer believes the claim is no longer valid among the literatures.

7. The fabrication process with laser-engraved microstructures may be very difficult to be repeated with high consistence for use of a low-cost CO2 laser, which always fluctuate on its energy output. Therefore, how to turn this fabrication process and the device into a practical use can be challenging.

8. Also, all the proposed applications have been investigated in multiple prior arts. For instance, the detection of the pulse waveforms from fingertip (Adv. Healthcare Mater. 2020, 2001023; ACS applied materials & interfaces, 2019, 11: 46399; Advanced Materials, 2022, 34: 2109357), the weighting or measuring tiny pressure change under high load (Nature Communications, 2020, 11: 209), and the human-robotic hand interaction (Nature Communications, 2014, 5:5747) have all been demonstrated before. It is not clear whether the proposed sensor structure offers better performance in any of the proposed applications.

9. The initial contact area of different structures for FEA shown in Fig.3 is estimated based on the SEM images according to the revision. However, this value should be acquired from the theoretical calculation because the contact area obtained from the SEM image is highly related to the preparation of the micro-structures. Furthermore, it is highly challenging to accurately measure the contact area of the rough interface using SEM, the author should also represent a detailed description on how to obtain the results in Methods section.

10. At last, some key references are still missing related to the iontronic pressure sensors in the field, for instance, the topical reviews Adv. Mater. 2021, 33, 2003464 and Adv. Funct. Mater. 2022, 32, 2110417.

Reviewer #4 (Remarks to the Author):

Revised paper may be accepted.

Reviewer #1:

The issues were well addressed. The manuscript can be published in NC.

Our response: We highly appreciate the reviewer's insightful comments that help significantly improve the overall quality of this work.

Our modification to the manuscript: None.

Reviewer #2:

The revised version of manuscript NCOMMS-22-32774A entitled “Iontronic pressure sensor with high sensitivity over ultra-broad linear range enabled by laser-induced gradient micro-pyramids” by Yang et al., addresses most of my previous concerns and is substantially improved over the previous version.

Our response: We highly appreciate the reviewer’s insightful comments that help significantly improve the overall quality of this work.

1. I still disagree in the use of low-cost to qualify these sensors. The authors try to justify claims of low cost by introducing Supplementary Table 1, which compares wet-etched silicon wafer and laser ablated PMMA. While I appreciate the author’s efforts, the comparison is mostly qualitative and not supported by references or other objective data. A true estimation of cost will require cost of ownership of equipment, operation, depreciation, in addition to throughput and yield calculations, noting that lithography is a high throughput parallel process while laser etching is based on a low throughput raster process. I do not think that claiming “low-cost” (I would frame it as: “reduced cost compared with...”) is necessarily wrong, but is clearly not supported by objective data and a clear methodology to estimate it. In my view, is also not necessary given the advances reported in this paper.

Our response: We thank the reviewer’s comment.

1. We have made the following changes to tone-down low-cost claims.

(1) We have deleted “low-cost” in the **Abstract**.

(2) We have changed “ a facile, effective, and low-cost” to “an effective and efficient” in the 3rd paragraph of the **Introduction**.

(3) We have deleted “low-cost” in the 4th paragraph of the **Introduction** and rephrased it as “reducing the cost and process complexity compared with photolithography” as suggested by the reviewer.

(4) We have updated Figure 1b and “Structure fabrication cost (from high to low)” has been changed into “prototyping efficiency (from low to high)”.

2. We have also included more microstructure template fabrication methods (with relevant references cited) in Supplementary Table 1, which compares “Materials”, “Facilities”, “Fabrication process”, “Environment requirement” and Notes (e.g., pros and cons).

3. We agree that lithography is a high throughput parallel process for the production of optimized structures, but laser ablation PMMA is more effective and efficient for prototyping to rapidly identify optimized structures. We also would like to provide more details about laser fabrication. In the laser printing system, we used various colors to represent different parameters (powers and speeds). Therefore, the laser-printing process depends on the number of colors in the whole pattern rather than pattern units. For example, the laser will first complete all the patterns in black and then red, and finally purple for the fabrication of GPMS₅₀₀ (Supplementary Table 3). As a result, the creation of an array becomes much faster. For instance, the fabrication of 17 microstructured pattern units in the array on a 3 inches PMMA board with the laser system is only 4.5 times (rather than 17 times) of that for the single unit (9’37” in Figure a and 43’2” in Figure b). Additionally, the speed parameter of the laser ablation also controls the fabrication running time. For the laser ablation process, higher laser power and faster laser speed result in the same performance as lower laser power and lower laser speed¹. As laser speed and power are increased by

3 times, the running time is reduced by more than half (Figure c). It is worth noting that laser equipment with higher power and faster speed can also be used to further reduce the fabrication time.

Comparison in the laser fabrication time for an array of pattern units with different laser parameters.

a Laser fabrication time of the single pattern unit. **b** Laser fabrication time of 17 pattern units in the array. **c** Comparison in the laser fabrication time between present laser parameters (left) and those with increased speed and power setting (right).

We have also tested a 3D-printing device (DREMEL3D45) to prepare different templates. The creation of a two-unit template (unit 2) takes twice as much time as the one-unit template (unit 1) (see the following plot).

Our modification to the manuscript:

1. We have updated the manuscript to tone down the claims of “low-cost”.

Abstract:

(1) “Despite the extensive developments of flexible capacitive pressure sensors, it is still elusive to simultaneously achieve excellent linearity over a broad pressure range, high sensitivity, and ultrahigh pressure resolution under large pressure preloads. Here, we present a programmable fabrication method for microstructures to integrate an ultrathin ionic layer.”

Introduction:

(2) “Therefore, there is an urgent demand for an effective and efficient fabrication method to prepare the programmable microstructure designed by quantitative numerical analysis.”

(3) “This work explores CO₂ laser with a Gaussian beam to fabricate programmable structures such as gradient pyramidal microstructures (GPM) for iontronic sensors, reducing the cost and process complexity compared with photolithography.”

Fig. 1b

(4)

2. We have updated Supplementary Table 1.

Supplementary Table 1. Different fabrication methods comparison for microstructure templates

Method	Materials	Facilities	Fabrication process	Environment requirement	Notes
Photolithography silica template ^{2,3}	 1. Silicon wafer 2. Photoresist 3. Silicon nitride 4. Potassium hydroxide 5. DI water 6. Trichloro-(1H,1H,2H,2H-perfluorooctyl) silane 7. IPA 8. Acetone 	 1. Cleanroom 2. Lithography equipment 3. Spin coater 4. Wet bench 5. O₂ plasma 	 1. Silicon nitride deposition 2. Mask preparation 3. Silicon etching 4. Remove the remaining silicon oxide 5. O₂ plasma treatment 6. Trichloro-(1H,1H,2H,2H-perfluorooctyl) silane deposition 	High temperature for etching	 1. Templates are regular, uniform, and high precision, along with complex fabrication processes. 2. High equipment requirements. 3. The template fabrication time > 5 h
3D printing ^{4,5}	Special materials for 3D printers (e.g., ABS,	3D printer	Printing (3D modeling needed)	Ambient conditions	1. Additive manufacture (low throughput)

	PLA, etc., depending on the type of printers)				2. Some materials (e.g. PLA) can not bear high temperatures (e.g., more than 40 °C) ⁴ , taking longer time to cure polymer.
Electrochemical oxidation aluminum oxide template ^{6, 7, 8}	 1. Aluminum plates 2. Perchloride acid 3. Ethanol 4. Oxalic acid 5. Phosphoric acid 6. Chromic acid 7. Trimethylchlorosilane gas 	 1. Voltage supply device 2. Temperature control device 	 1. Annealing 2. Electrochemical polishing 3. Two-step anodization 4. Surface treatment 	 1. High temperature (annealing and chemical etching) 2. Low temperature (anodization) 3. DC voltage 	 1. Structure limitation (most of them are templates for cone and column). 2. High density and uniform parallel nanopores.
Micro-engraving ^{9, 10}	Plastic boards (e.g., polycarbonate)	Engraving machine and microdrill	 1. Cutter switching 2. Engraving 	Ambient conditions	1. The shape and size of microstructures are limited by cutters.

					2. Cutters are easily getting abraded.
Laser ablation	PMMA boards	CO ₂ laser system	Printing	Ambient conditions	 1. Easy to start. 2. High prototyping efficiency. 3. High design flexibility to adjust the size/shape/height/density of microstructures.

2. On the Introduction, page 2, revise grammar following text: “Ionic liquids (IL) of atomically thin (~ 1 nm) between positive and negative charges....” Perhaps “Atomically thin ionic liquid layers (~ 1 nm)....”

Our response: We thank the reviewer for the suggestion and we have updated the sentence on page 2.

Our modification to the manuscript: “Atomically thin ionic liquids (ILs) layers (~1 nm) between positive and negative charges in the electron double layer (EDL) at the electrode/dielectric interface can significantly increase the piezo-capacitive effect²⁵.”

3. On page 3, replace the word "ingenious" with "various" or "novel" in the phrase: “Efforts to address this challenge lead to the exploration of ingenious structure designs in iontronic sensors.”

Our response: We thank the reviewer for the suggestion and we have updated the sentence.

Our modification to the manuscript: We have changed “ingenious” to “various”. “Efforts to address this challenge lead to the exploration of various structure designs in iontronic sensors.”

4. On page 9, replace “Supplementary Fig. 4de” with: “Supplementary Fig. 4d-e”

Our response: We thank the reviewer for the suggestion and we have updated the sentence.

Our modification to the manuscript: We have changed “Supplementary Fig. 4de” to “Supplementary Fig. 4d-e”.

“..., different sizes of pyramidal microstructures can be formed (Supplementary Fig. 4d-e).”

Reviewer #3:

Although the revised manuscript has responded to many critical comments in the revised form, the fundamental issues presented in the current form are still not fully addressed. Particularly, the mechanism to achieve outstanding performance (e.g., high linearity and high performance) is not well explained; the low-cost processing remains unclear or not convinced; the high sensitivity and SNR results are lack of critical information; the fabrication process with laser-engraved microstructures may be very difficult to be repeated with high consistence to satisfy real high-performance medical needs; and at last, the applications proposed have been widely investigated in multiple prior arts. Therefore, it still has not yet reached the quality of innovation and performance criteria of Nature Communications in its revised form. The detailed comments have been included below:

Our response: We appreciate the reviewer's comments and we have updated the manuscript to address them as detailed in the following sections.

1. We have updated the theoretical analysis of gradient microstructures to help explain the performance parameters (comment #1).
2. We have updated the discussion to tune down the low-cost claims (comment #2).
3. We have included details to help support the high sensitivity (i.e., initial small contact area and the formation of the EDL) and high SNR results (comment #5) in the revised manuscript.
4. The consistency in the fabricated structure for the target applications has been illustrated (comment #7).
5. We have updated the manuscript to indicate that the applications were used to demonstrate the performance of the developed sensors (comment #8).

1. The main concept of height-gradient structure to increase the device linearity presented is quite similar to ref. 9 in this manuscript, which uses gradient micro-dome array to achieve high linearity

in a capacitive counterpart. Unfortunately, this paper does not offer a comprehensive comparison on its performance with that of ref.9; in particular, the shape influence of the microstructure vs the device performance. Moreover, the theory behind its mechanism presented in the rebuttal version is not highly relevant to its performance. Specifically, in A4 of the rebuttal letter, the claimed linear relationship between capacitive output of the sensor and the applied pressure cannot be derived from Eq. 3 and Eq. 4. Moreover, it is confusing that the proposed equations in the revision do not explain/predict the performance, including linearity range, sensitivity, etc.

Our response: We appreciate the reviewer's comments.

1. Despite the inspiration from Ref. 9, there are several differences between Ref. 9 and this work (the superscript in this section denotes the reference in the manuscript):

(1) The comprehensive comparison has been provided in Supplementary Table 2 (Num. 20).

(2) We mentioned the influence of structure shape on pressure sensors in the **Introduction** by citing ref.9: “Among varying microstructures such as micro-dome⁹ and hierarchical fabrics¹⁰, the pyramidal structures^{11,12} exhibit excellent compressibility and progressive deformation with increasing pressure to result in high sensitivity in a large linear range.”;

(3) We mentioned the effects of IL layers on capacitors in the 2nd paragraph of the **Introduction**: “Atomically thin ionic liquids (ILs) layers (~1 nm) between positive and negative charges in the electron double layer (EDL)²⁵ at the electrode/dielectric interface can significantly increase the piezo-capacitive effect²⁶.”

(4) Ref. 9 used high-k PDMS/CNT composite as the dielectric material, so the changes in capacitance only depend on the contact area at the electric/dielectric interface. However, without EDL at the dielectric/electrode interface, the sensitivity of the resulting sensor is low (0.065 kPa⁻¹).

(5) We emphasized the current gap in the 3rd paragraph of the **Introduction** by citing Ref. 9: “High linearity over a broad sensing range usually comes at a cost of reduced sensitivity (e.g., 9280 kPa⁻¹ within 20 kPa³¹ vs. 0.065 kPa⁻¹ within 1700 kPa⁹).”

(6) We demonstrated the use of various laser patterns to create different microstructures such as the microdome-like structure (or conical frustums in Supplementary Fig. 2b). The mechanical simulation results (Fig. 3b) helped validate the trend of changes for different shapes in the experiment (Fig. 3c). We also studied the influence of the electric field on the performance of the sensor (Fig. 4).

2. We have updated the manuscript to indicate that the theoretical analysis was used to help understand the sensor’s performance without bending. In addition, we have added a schematic diagram (Supplementary Fig. 9c) to explain the performance change of the sensor under bending. Pyramids distributed on the sides are less pressed under bending, which helps explain the reduced the gradient microstructures-extended sensing range. However, the contact to the gradient microstructures upon increasing pressure is still from high to low pyramids, which provides a small initial contact area and a gradually increased contact area. Therefore, the sensors can still exhibit a relatively large linear sensing range.

3. The deformation of the pyramid and changes in the contact area changes under pressure loadings are more complex than previous studies (e.g., compared with the

hemispherical microstructures used in Ref. 9). Although FEA demonstrated the process to help optimize sensitivity and linearity of the sensor (Figs. 3 and 4), the sensor performance affected by the mechanics and electric field increases the complexity of the problem. Therefore, the theoretical analysis in the current study is only focused on mechanical simulation to help understand the improvement of the sensitivity due to gradient structures.

As previously mentioned, the effective cross-section w_{eff} is given by

$$w_{\text{eff}} = \sum_{i=1}^{N(F)} w_i(F) \propto \sum_{i=1}^{N(F)} \sqrt{F_i}. \quad (3)$$

Using Cauchy–Schwarz inequality,

$$w_{\text{eff}} \propto \sum_{i=1}^{N(F)} \sqrt{F_i} \leq \sqrt{\sum_{i=1}^{N(F)} F_i} \approx \sqrt{N(F)\langle F \rangle}. \quad (4)$$

Thereby, the effective capacitance C_{eff} is given by

$$C_{\text{eff}} \leq N(F)\langle F \rangle, \quad (5)$$

where $N(F)$ is the number of pillars deformed with force $\pi_N < F < \pi_{N+1}$ and average force $\frac{1}{N(F)} \sum_{i=1}^{N(F)} F_i = \langle F \rangle$. The number of pillars $N(F)$ increases monotonically with force as the force increases with the number of deformed pillars. Moreover, the average force $\langle F \rangle$ corresponds to the pressure applied P over the sensor cross-sectional area A . Therefore, the number of pillars $N(F)$ is also a function of the applied pressure $N(P)$.

The slope of effective capacitance C_{eff} with pressure or sensitivity S is thus given by

$$S = \frac{\partial C_{\text{eff}}}{\partial P} \leq kAN(P), \quad (5)$$

where k is the proportional constant. As the effective capacitance depends on the square of the effective cross-section, the slope of the capacitance-force is upper-bounded with the number of pillars N , which depends on the pressure P . Although this theory gives a theoretical background to study the influence of gradient structures, it doesn't directly predict sensitivity and linear range, which will be our focus in future

works. The combination of this theory with bending deformations is out of the scope of this paper and will be pursued in our future studies.

Our modification to the manuscript:

1. We have explained the differences between Ref. 9 and this work.
2. We have updated the description of the sensor's performance under bending conditions.

The flexibility of the sensor allows it to easily attach to a curved surface (Supplementary Fig. 9a). The outstanding performance of the pressure sensor (e.g., high sensitivity and linearity) usually benefits from ingenious structure designs. However, the sensor performance is often significantly reduced under bending conditions³³ likely due to the change in the microstructures. As the bending radius is decreased from 45 to 31 and then to 13 mm, the sensitivity of GPML₇₀₀ decreases from 11.96 to 6.10 and then to 1.86 kPa⁻¹ (Supplementary Fig. 9b). This decrease results from the increased initial contact area and the reduced gap between the structured bottom electrode and the dielectric layer, increasing the initial capacitance (before pressure loading). The increased initial capacitance reduces the capacitance changes of the sensor under pressure and thus the sensitivity. Additionally, the non-uniform contact to the gradient pyramids upon bending reduces the sensing range from 1700 kPa to ca. 500 kPa (Supplementary Fig. 9c). However, the iontronic pressure sensor still exhibits reasonably good sensitivity in a relatively large linear sensing range due to the small initial contact area and gradually increased contact area with the increasing pressure. Further improvement can be possible when the iontronic pressure sensor is directly fabricated on the 3D freeform surfaces with a modified laser setup⁵⁷.

3. We have updated the theoretical analysis.

Note 2. Theoretical analysis of UPM and GPM at the dielectric/electrode interface without bending

The normalized cross-section w of the contact surface of the pyramid microstructure increases proportionally to the square root of the compression force F against the pyramid¹¹.

$$w \propto \sqrt{F}. \quad (1)$$

However, the capacitance C is directly proportional to the area or square of the cross-section of the contact surface, thus, the capacitance becomes directly proportional to the compressive force.

$$C \propto w^2 \propto F. \quad (2)$$

The linear dependence of capacitance and force thus originates from the non-linear relationship between the cross-section and compressive force given in equation (1). For gradient microstructure, the effective cross-section w_{eff} of the contact increases in a cascading order for each new pillar $i \leq N$ after exceeding every corresponding force π_i . The corresponding force π_i depends on the gradient of the microstructure, as the gradient increases, more exceeding force π_i is required to start deformation of the i th pillar. For example, the corresponding force will be zero for all the pillars for a uniform pillar distribution, whereas, for a gradient pillar distribution, only the exceeding force π_1 corresponding to the initially deformed pillar $i = 1$ will be zero, and this force increases with the pillar index $\pi_i < \pi_{i+1} \forall i \leq N$. The force F_i deforms each pillar i with width w_i as the following relationship

$$w_i(F) = \begin{cases} w_i \propto \sqrt{F_i}, & F > \pi_i \\ 0, & F \leq \pi_i \end{cases}$$

The effective cross-section w_{eff} is given by

$$w_{\text{eff}} = \sum_{i=1}^{N(F)} w_i(F) \propto \sum_{i=1}^{N(F)} \sqrt{F_i}. \quad (3)$$

Using Cauchy–Schwarz inequality,

$$w_{\text{eff}} \propto \sum_{i=1}^{N(F)} \sqrt{F_i} \leq \sqrt{\sum_{i=1}^{N(F)} F_i} \approx \sqrt{N(F)\langle F \rangle}. \quad (4)$$

Thereby, the effective capacitance C_{eff} is given by

$$C_{\text{eff}} \leq N(F)\langle F \rangle, \quad (5)$$

where $N(F)$ is the number of pillars deformed with force $\pi_N < F < \pi_{N+1}$ and average force $\frac{1}{N(F)} \sum_{i=1}^{N(F)} F_i = \langle F \rangle$. The number of pillars $N(F)$ increases

monotonically with force as the force increases with the number of pillars deformed.

Moreover, the average force $\langle F \rangle$ corresponds to the pressure applied P over the sensor cross sectional area A . Therefore, the number of pillars $N(F)$ is also a function of pressure applied $N(P)$.

The slope of effective capacitance C_{eff} with pressure or sensitivity S is thus given by

$$S = \frac{\partial C_{\text{eff}}}{\partial P} \leq kAN(P), \quad (5)$$

where k is the proportional constant. As the effective capacitance depends on the square of the effective cross-section, the slope of the capacitance force curve is upper-bounded by the number of pillars N that depends on the pressure P .

2. Secondly, the claimed low-cost feature is not very solid in the practical view. Particularly, in A1 of the rebuttal letter, the authors have only compared the cost of the sensors with that prepared by lithography process and acknowledged as a high-cost approach. The comparisons with the other low-cost processes, for instance, molding, printing, etc. have been largely ignored. However, the proposed sensor structure in this manuscript involves a sputtered gold layer as the electrode, which is normally considered as an expensive step, in comparison with the screen-printing or electroplated process with a much lower cost of metal compounds.

Our response: We appreciate the reviewer's comments. Although we used the benchtop sputter coater (often used for SEM sample preparation; much cheaper than the conventional sputters), we have updated the discussion to tune down the low-cost claims. We also agree that the cost of the electroplated and screen-printed gold may be leveraged to further reduce the cost.

1. We have updated the discussion to tune down the low-cost claims.

(1) We have deleted "low-cost" in the **Abstract**.

(2) We have changed "a facile, effective, and low-cost" to "an effective and efficient" in the 3rd paragraph of the **Introduction**.

(3) We have deleted "low-cost" in the 4th paragraph of the **Introduction** and rephrased it as "reducing the cost and process complexity compared with photolithography" as suggested by the reviewer.

(4) We have updated Figure 1b and "Structure fabrication cost (from high to low)" has been changed into "prototyping efficiency (from low to high)".

2. We have also included more microstructure template fabrication methods (with relevant references cited) in Supplementary Table 1, which compares "Materials", "Facilities", "Fabrication process", "Environment" and Notes (e.g., pros and cons).

Our modification to the manuscript:

1. We have updated the manuscript to tone down the low-cost claims.

Abstract:

(1) "Despite the extensive developments of flexible capacitive pressure sensors, it is still elusive to simultaneously achieve excellent linearity over a broad pressure range, high sensitivity, and ultrahigh pressure resolution under large pressure preloads. Here, we present a programmable fabrication method for microstructures to integrate an ultrathin ionic layer."

Introduction:

(2) “Therefore, there is an urgent demand for an effective and efficient fabrication method to prepare the programmable microstructure designed by quantitative numerical analysis.”

(3) “This work explores CO₂ laser with a Gaussian beam to fabricate programmable structures such as gradient pyramidal microstructures (GPM) for iontronic sensors, reducing the cost and process complexity compared with photolithography.”

Fig. 1b

(4)

2. Supplementary Table 1 has been updated with more template fabrication methods. There are two methods for 3D printing to create microstructure: one is to directly print the structure itself and the other is to print a 3D template. Here, we only discuss template printing because the other one is not relevant.

Supplementary Table 1. Different fabrication methods comparison for microstructure templates

Method	Materials	Facilities	Fabrication process	Environment requirement	Notes
Photolithography silica template ^{2,3}	 1. Silicon wafer 2. Photoresist 3. Silicon nitride 4. Potassium hydroxide 5. DI water 6. Trichloro-(1H,1H,2H,2H-perfluorooctyl) silane 7. IPA 8. Acetone 	 1. Cleanroom 2. Lithography equipment 3. Spin coater 4. Wet bench 5. O₂ plasma 	 1. Silicon nitride deposition 2. Mask preparation 3. Silicon etching 4. Remove the remaining silicon oxide 5. O₂ plasma treatment 6. Trichloro-(1H,1H,2H,2H-perfluorooctyl) silane deposition 	High temperature for etching	 1. Templates are regular, uniform, and high precision, along with complex fabrication processes. 2. High equipment requirements. 3. The template fabrication time > 5 h
3D printing ^{4,5}	Special materials for 3D printers (e.g., ABS,	3D printer	Printing (3D modeling needed)	Ambient conditions	1. Additive manufacture (low throughput)

	PLA, etc., depending on the type of printers)				2. Some materials (e.g. PLA) can not bear high temperatures (e.g., more than 40 °C) ⁴ , taking longer time to cure polymer
Electrochemical oxidation aluminum oxide template ^{6, 7, 8}	 1. Aluminum plates 2. Perchloride acid 3. Ethanol 4. Oxalic acid 5. Phosphoric acid 6. Chromic acid 7. Trimethylchlorosilane gas 	 1. Voltage supply device 2. Temperature control device 	 1. Annealing 2. Electrochemical polishing 3. Two-step anodization 4. Surface treatment 	 1. High temperature (annealing and chemical etching) 2. Low temperature (anodization) 3. DC voltage 	 1. Structure limitation (most of them are templates for cone and column). 2. High density and uniform parallel nanopores.
Micro-engraving ^{9, 10}	Plastic boards (e.g., polycarbonate)	Engraving machine and microdrill	 1. Cutter switching 2. Engraving 	Ambient conditions	 1. The shape and size of microstructures are limited by cutters. 2. Cutters are easily getting abraded.

Laser ablation	PMMA boards	CO ₂ laser system	Printing	Ambient conditions	1. Easy to start.2. High prototyping efficiency.3. High design flexibility to adjust the size/shape/height/density of microstructures.
----------------	-------------	------------------------------	----------	--------------------	--

3. It is highly skeptical on the high-performance demonstration of the sensor in terms of the resolution, in which a value of 0.00725% under ultrahigh pressure of 2000 kPa was announced. However, the ballistocardiograph signals of a typically human subject is not observed (similar to a pulse wave signal), which is quite surprising, because the magnitude of the signals should be considerably greater than that of the sensor resolution claimed in the manuscript.

Our response: We appreciate the reviewer's comments. We have recorded a video (Supplementary Movie 3) to show the process of dropping the pen. The optical image of the sensor placed under the chair's leg is attached to the bottom right corner of the video. The subject (weight of 79.5 kg, ca. 10 kg heavier than the one shown in the previous manuscript) sitting still on the chair with back relaxed. The ballistocardiograph (BCG) signal is important in biomedicine, but complex signal postprocessing (e.g., amplification, band-pass filter¹², etc.) is often needed to obtain the accurate waveform as it is more susceptible to changes in the environmental conditions and motion artifacts¹³ (e.g., breath, environment noise, etc.). Without the relevant equipment and expertise for BCG, we could not provide the relevant demonstrations, which will be pursued in further studies.

Our modification to the manuscript: We have provided a video (Supplementary Movie 3) to show the process of dropping the pen.

4. In A8, the author shows interesting measurement results to load and unload an object with 50g in several milliseconds. It is extremely difficult to conduct such an experiment to the reviewer's knowledge. A detailed description of the test, including a video clip, should be presented to demonstrate such a weight-loading experiment.

Our response: We appreciate the reviewer's comments. The real-time measurements with the LCR meter (Hioki, IM3536 in "Fast Mode" Overview of IM3536) and upgraded software are recorded with 3 decimal places in a second. The time increment between two recorded signals varies in every experiment, but it is in the range from 0.001 to 0.005 s. The response/recovery time is also affected by the placement and removal speed of the weight. We have uploaded Supplementary Movie 4 to show the test process.

Our modification to the manuscript: We have provided Supplementary Movie 4 to show the test process.

5. In A9, the results presented in the revised version show a significant improvement on the device SNR from its original article, however, it is not well explained how the conducted experiment differs from the original (in Fig. 5c), why does it now show significantly better outcomes in SNR in the revision.

Our response: We appreciate the reviewer's reminder. As the highest micropiramids are located at the four corners of the sensing unit, we changed the small soft tissue paper into a larger weighing paper that can cover the entire sensing unit. Therefore, the tiny pressure can be captured. We have uploaded Supplementary Movie 5 to show the test process.

Our modification to the manuscript: We have changed the "tissue" into "weighing paper" and updated the picture in Fig. 5c.

1. “The initially small contact area at the fine tip of the pyramid created from the laser-ablated rough surface (Supplementary Fig. 10a) further provides the iontronic pressure sensor with a low limit detection of 0.36 Pa to detect a light weighing paper (Fig. 5c).”

2. **Fig. 5c** caption: “Demonstration of the detection limit of 0.36 Pa with repeated loading of a small piece of weighing paper on the sensor.”

6. In the application of “High pressure resolution under larger pressure preloads”, the calculated pressure change of dropping pens is incorrect. The result based on its data should yield 145Pa, but not 14.5Pa, leading to the flawed conclusion that “The extraordinarily high pressure resolution of 0.00725% under a huge pressure load of 2 MPa significantly outperforms the commercial weight scale (Supplementary Fig. 10c), human skin (~ 7%), and other microstructured pressure sensors previously reported in the literature”. If corrected, the reviewer believes the claim is no longer valid among the literatures.

Our response: We thank the reviewer for the careful check. We have corrected the typo and changed “14.5” to “145” Pa. The pen (5.8 g) and preload (a person and a chair of 80 kg) are applied on the same area (4 cm²), so the weight ratio of the two is 0.0000725. Therefore, the conclusion “The extraordinarily high pressure resolution of 0.00725% under a huge pressure load of 2 MPa significantly outperforms the

commercial weight scale (Supplementary Fig. 10c), human skin (~ 7%), and other microstructured pressure sensors previously reported in the literature” is still valid.

Our modification to the manuscript: We have changed 14.5 into 145 in the revised manuscript.

1. “Demonstration of high-pressure resolution with additional tiny pressure of 145 Pa detected under a large preload of 2 MPa.”
2. “The additional tiny pressure under a large preload can also be accurately captured with a person (69.4 kg) holding a brush pot on the chair (preload pressure of ~2000 kPa) to catch dropping pens (each of 5.8 g, pressure of ~145 Pa) in sequence.”
3. “Additionally, the fine features on the GPM surface provide minimized initial contact area for the sensor to detect an ultralow pressure of 0.36 Pa without preload and tiny pressure variation of 145 Pa with a pressure preload of 2 MPa for an ultrahigh high pressure resolution of 0.00725%.”

7. The fabrication process with laser-engraved microstructures may be very difficult to be repeated with high consistence for use of a low-cost CO2 laser, which always fluctuate on its energy output. Therefore, how to turn this fabrication process and the device into a practical use can be challenging.

Our response: We thank the reviewer’s comments.

1. First of all, the fabrication of pressure sensors with simple processes in the previous reports still relies on replicating microstructures from objects (e.g., sandpaper¹⁴, plant^{15, 16}, etc.), directly using microstructures (e.g., textile fabric¹⁷, etc.) of objects, or using relative random microstructures (e.g., electrospinning enabled fiber network¹⁸, sugar¹⁹ particle dissolution enabled porous structure, hot-air-gun induced quasi-

hemisphere²⁰, nanoparticle dispersity enabled surface microstructure²¹, magnetic force induced micro cilia²²). Compared with these methods, the microstructures created by the CO₂ laser system with reasonably consistent processing quality (the official website of the laser system: Excellent Laser Cutting, Engraving, and Marking Quality) are more consistent.

2. We have fabricated two PMMA templates by using the same laser parameters (e.g., power, speed, PPI, etc.) and the resulting microstructures obtained from these two templates show reasonably good consistency in the morphology (e.g., height, outline, surface topography, etc).

3. Finally, we agree that the accuracy of the final microstructures may be affected by many factors, including equipment quality, operation, and variations in environmental conditions, among others. However, the differences can be easily accounted for by

sensor performance calibration after fabrication, which is relatively easy with the high linearity of the sensor.

Our modification to the manuscript: We have updated the Manuscript and the Supplementary information to discuss the consistency of laser fabrication.

1. Manuscript:

Methods

Fabrication of the PMMA mold:

“The comparison between microstructures obtained from two PMMA templates that were separately created by using the same laser parameters demonstrated the reasonably good consistency of laser fabrication (Supplementary Fig. 12).”

2. Supplementary information:

Supplementary Fig. 12. The comparison between microstructures obtained from two PMMA templates that were separately created by using the same laser parameters,

demonstrating reasonably good consistency in the morphology (e.g., height, outline, and surface topography).

8. Also, all the proposed applications have been investigated in multiple prior arts. For instance, the detection of the pulse waveforms from fingertip (Adv. Healthcare Mater. 2020, 2001023; ACS applied materials & interfaces, 2019, 11: 46399; Advanced Materials, 2022, 34: 2109357), the weighting or measuring tiny pressure change under high load (Nature Communications, 2020, 11: 209), and the human-robotic hand interaction (Nature Communications, 2014, 5:5747) have all been demonstrated before. It is not clear whether the proposed sensor structure offers better performance in any of the proposed applications.

Our response: We appreciate the reviewer's comments. The innovation of this work focuses on the method to rapidly fabricate and optimize the gradient microstructures in the pressure sensor with unique sensing performance parameters for these applications. This effort echoes the need for developments as pointed out in one of Prof. Zhenan Bao's recent review papers: "the more control over the structure, the greater the tunability of the sensor's performance" (Adv. Funct. Mater, 2020, 30, 2003491). The demonstrated applications in fingertip pulse and weighting measurements and robotic hand interaction were used to highlight the performance of the sensor in fast response/recovery time, high linearity, high pressure resolution, large sensing range, and low detection limit. In particular, previously reported fingertip pulse sensors (e.g., Adv. Healthcare Mater. 2020, 2001023; ACS applied materials & interfaces, 2019, 11: 46399) are often associated with a narrow sensing range for small pressure detection only (pulse and joint/skin motions)^{25, 26}. With structures replicated from sandpaper (Nature Communications, 2020, 11: 209), the sensor only provides a maximum sensing range of 360 kPa and the tiny pressure detection is also under the preload within this

range. In comparison, our sensor exhibits a maximum linear sensing range of 3000 kPa and the tiny pressure detection is also under the preload pressure of 2 MPa. Compared with the demonstration that uses a strain sensor array to detect different actions of the robotic hand (Nature Communications, 2014, 5:5747), we exploit a PID control system to provide accurate feedback.

Our modification to the manuscript: We have updated the manuscript to note that the use of demonstrated applications is to highlight the performance of the developed sensor.

Discussion:

“The potential of the flexible iontronic sensor with high performance is highlighted and showcased in the proof-of-concept demonstrations of fingertip pulse monitoring from human-machine interactions, high-performance smart chairs with ultrahigh pressure resolution, and intelligent robotic hands for dexterous object manipulation.”

9. The initial contact area of different structures for FEA shown in Fig.3 is estimated based on the SEM images according to the revision. However, this value should be acquired from the theoretical calculation because the contact area obtained from the SEM image is highly related to the preparation of the micro-structures. Furthermore, it is highly challenging to accurately measure the contact area of the rough interface using SEM, the author should also represent a detailed description on how to obtain the results in Methods section.

Our response: We appreciate the reviewer’s comments. We have used simulation to calculate the initial contact area (A_0) at the electrode/dielectric interface as reported previously (ACS Nano 2022, 16, 4338-4347).

Our modification to the manuscript: We have changed the calculation method for the initial contact area in **Methods** and updated the results in Fig. 3b.

“A pressure of 1600 kPa was applied on the top electrode that was treated as a rigid plate (not shown in Fig. 3a). The initial electrode/dielectric contact interface (A_0) was estimated to be 5 kPa.”

10. At last, some key references are still missing related to the iontronic pressure sensors in the field, for instance, the topical reviews *Adv. Mater.* 2021, 33, 2003464 and *Adv. Funct. Mater.* 2022, 32, 2110417.

Our response: We appreciate the reviewer’s reminder. We have cited the two papers in the **Introduction**.

Our modification to the manuscript:

1. We have cited the two references in the second paragraph of **Introduction**.

Atomically thin ionic liquids (ILs) layers (~1 nm) between positive and negative charges in the electron double layer (EDL)²⁵ at the electrode/dielectric interface can significantly increase the piezo-capacitive effect²⁶.

Ref 25: *Adv. Mater.* 2021, 33, 2003464

Ref 26: *Adv. Funct. Mater.* 2022, 32, 2110417

2. We have updated the references in Fig. 5a.

Reviewer #4:

Revised paper may be accepted.

Our response: We highly appreciate the reviewer's insightful comments that help significantly improve the overall quality of this work.

Our modification to the manuscript: None.

References:

1. Mishra NK, Patil N, Anas M, Zhao X, Wilhite BA, Green MJ. Highly selective laser-induced graphene (LIG)/polysulfone composite membrane for hydrogen purification. *Applied Materials Today* **22**, 100971 (2021).
2. Lee S, *et al.* A highly sensitive bending sensor based on controlled crack formation integrated with an energy harvesting pyramid layer. *Advanced Materials Technologies* **3**, 1800307 (2018).
3. Jung M, *et al.* Transparent and flexible mayan-pyramid-based pressure sensor using facile-transferred indium tin oxide for bimodal sensor applications. *Scientific reports* **9**, 14040 (2019).
4. Kang B, Hyeon J, So H. Facile microfabrication of 3-dimensional (3D) hydrophobic polymer surfaces using 3D printing technology. *Appl Surf Sci* **499**, 143733 (2020).
5. Zhuo B, Chen S, Zhao M, Guo X. High sensitivity flexible capacitive pressure sensor using polydimethylsiloxane elastomer dielectric layer micro-structured by 3-D printed mold. *IEEE Journal of the Electron Devices Society* **5**, 219-223 (2017).
6. Hoang XT, Nguyen DT, Dong BC, Nguyen HN. Fabrication of carbon nanostructures from polymeric precursor by using anodic aluminum oxide (AAO) nanotemplates. *Advances in Natural Sciences: Nanoscience and Nanotechnology* **4**, 035013 (2013).
7. Dudem B, Ko YH, Leem JW, Lee SH, Yu JS. Highly transparent and flexible triboelectric nanogenerators with subwavelength-architected polydimethylsiloxane by a nanoporous anodic aluminum oxide template. *ACS applied materials & interfaces* **7**, 20520-20529 (2015).
8. Hoa ND, Van Quy N, Cho Y, Kim D. An ammonia gas sensor based on non-catalytically synthesized carbon nanotubes on an anodic aluminum oxide template. *Sensors and Actuators B: Chemical* **127**, 447-454 (2007).
9. Zhou Q, *et al.* Tilted magnetic micropillars enabled dual-mode sensor for tactile/touchless perceptions. *Nano Energy* **78**, 105382 (2020).

10. Ji B, *et al.* Gradient Architecture - Enabled Capacitive Tactile Sensor with High Sensitivity and Ultrabroad Linearity Range. *Small* **17**, 2103312 (2021).
11. Liu S, Olvera de la Cruz M. Deformation of elastomeric pyramid pen arrays in cantilever - free scanning probe lithography. *Journal of Polymer Science Part B: Polymer Physics* **56**, 731-738 (2018).
12. Zhou Z, *et al.* Single-layered ultra-soft washable smart textiles for all-around ballistocardiograph, respiration, and posture monitoring during sleep. *Biosensors and Bioelectronics* **155**, 112064 (2020).
13. Lu H, Zhang H, Lin Z, Huat NS. A novel deep learning based neural network for heartbeat detection in ballistocardiograph. In: *2018 40th Annual International Conference of the IEEE Engineering in Medicine and Biology Society (EMBC)*. IEEE (2018).
14. Tang X, *et al.* Multilevel microstructured flexible pressure sensors with ultrahigh sensitivity and ultrawide pressure range for versatile electronic skins. *Small* **15**, 1804559 (2019).
15. Liu Y-Q, Zhang J-R, Han D-D, Zhang Y-L, Sun H-B. Versatile electronic skins with biomimetic micronanostructures fabricated using natural reed leaves as templates. *ACS applied materials & interfaces* **11**, 38084-38091 (2019).
16. Liu Z, *et al.* Natural bamboo leaves as dielectric layers for flexible capacitive pressure sensors with adjustable sensitivity and a broad detection range. *RSC Advances* **11**, 17291-17300 (2021).
17. Atalay O, Atalay A, Gafford J, Walsh C. A highly sensitive capacitive - based soft pressure sensor based on a conductive fabric and a microporous dielectric layer. *Advanced materials technologies* **3**, 1700237 (2018).
18. Li S, Li R, González OG, Chen T, Xiao X. Highly sensitive and flexible piezoresistive sensor based on c-MWCNTs decorated TPU electrospun fibrous network for human motion detection. *Compos Sci Technol* **203**, 108617 (2021).

19. Peng S, Wu S, Yu Y, Xia B, Lovell NH, Wang CH. Multimodal Capacitive and Piezoresistive Sensor for Simultaneous Measurement of Multiple Forces. *ACS Appl Mater Interfaces* **12**, 22179-22190 (2020).
20. Zhang Y, *et al.* Multifunctional interlocked e-skin based on elastic micropattern array facilely prepared by hot-air-gun. *Chem Eng J* **407**, 127960 (2021).
21. Kim H, *et al.* Transparent, flexible, conformal capacitive pressure sensors with nanoparticles. *Small* **14**, 1703432 (2018).
22. Zhou Q, *et al.* A bio-inspired cilia array as the dielectric layer for flexible capacitive pressure sensors with high sensitivity and a broad detection range. *Journal of Materials Chemistry A* **7**, 27334-27346 (2019).
23. Meng K, *et al.* Ultrasensitive fingertip-contacted pressure sensors to enable continuous measurement of epidermal pulse waves on ubiquitous object surfaces. *ACS applied materials & interfaces* **11**, 46399-46407 (2019).
24. Lin Q, *et al.* Highly sensitive flexible iontronic pressure sensor for fingertip pulse monitoring. *Advanced Healthcare Materials* **9**, 2001023 (2020).
25. Zhu P, *et al.* Skin-electrode iontronic interface for mechanosensing. *Nat Commun* **12**, 4731 (2021).
26. Das PS, Chhetry A, Maharjan P, Rasel MS, Park JY. A laser ablated graphene-based flexible self-powered pressure sensor for human gestures and finger pulse monitoring. *Nano Research* **12**, 1789-1795 (2019).

REVIEWER COMMENTS

Reviewer #2 (Remarks to the Author):

I thank the authors for the changes made. In the revised manuscript they have answered all of my concerns. I can now support the publication of this manuscript in Nature Communications.

A final suggestion to improve readability:

Line 78: Replace: "This work explores CO2 laser with a Gaussian beam..." with : "This work explores the use of a CO2 laser with a Gaussian beam profile"

Reviewer #3 (Remarks to the Author):

In the revised manuscript, authors have attempted to respond to majority of the critical comments, however, several key issues, including its theoretical model, claimed superhigh resolution, as well as experimental characterization of the time response, still need to be addressed in an appropriate way before its acceptance. The detailed comments are included as follow:

1. In A1, the revised theoretic analysis for sensor performance appears to be irrelevant to the device characterization. It mainly links the device sensitivity to the number of pillars, not to the cause of high linearity nor any other claimed performance of the sensor itself.
2. In A1, the influence of bending to the performance of the pyramids structure should be conducted through experiments or citing the theory from existing literatures.
3. In A3, though the resolution of the sensor was calculated to be 0.0075%, the practical resolution seemed to be considerably poorer than that of the regular commercial weight scale. Thus, to compare the resolution/sensitivity of the proposed device with the commercial scale, the proposed measurement is inadequate. Even from a pressure sensor/load cell/scale with a reasonable sensitivity, the BCG signals can be easily recorded by using an LCR meter without additional filtering or signal processing capacity. Specifically, a number of prior arts have shown that the commercial scales, with a resolution of 0.1%, can record the typical BCG signals (e.g., Fig 6d) as well as respiratory patterns from human subjects (e.g., *Nutrients* 2022, 14, 2552. <https://doi.org/10.3390/nu14122552>, *IEEE Journal of Biomedical and Health Informatics*, vol. 24, no. 1, pp. 69-78), which has not been observed in the manuscript nor in the attached video. Therefore, the claim that the resolution of the sensor outperforms that of the commercial weight scales may not be appropriate.
4. In A4, the method to loading and unloading using hand during the response time test could lead to highly unreliable or unrepeatable results. Thus, it is highly recommended that authors follow the standard practice in the published literatures, from which reliable data can be yielded (e.g., *Nature Communications*, 2014, 5: 3132).

Reviewer #2:

1. *I thank the authors for the changes made. In the revised manuscript they have answered all of my concerns. I can now support the publication of this manuscript in Nature Communications.*

Our response: We highly appreciate the reviewer's insightful comments that help significantly improve the overall quality of this work.

2. *A final suggestion to improve readability:*

Line 78: Replace: "This work explores CO2 laser with a Gaussian beam..." with : "This work explores the use of a CO2 laser with a Gaussian beam profile"

Our response: We thank the reviewer's careful check and we have updated the corresponding sentence.

Our modification to the manuscript: This work explores the use of a CO₂ laser with a Gaussian beam profile to fabricate programmable structures...

Reviewer #3:

In the revised manuscript, authors have attempted to respond to the majority of the critical comments, however, several key issues, including its theoretical model, claimed superhigh resolution, as well as experimental characterization of the time response, still need to be addressed in an appropriate way before its acceptance. The detailed comments are included as follow:

Our response: We highly appreciate the reviewer's insightful comments that help significantly improve the overall quality of this work.

Our modification to the manuscript: We have updated the Manuscript and Supplementary Information (with a detailed response provided in the following response to the reviewer's comments).

1. The limitation of the current methodology has been added to the **Discussion**.
2. The description of the pressure resolution has been updated.
3. Supplementary Movie 6 has been uploaded to verify the response of the sensor.

1. In A1, the revised theoretic analysis for sensor performance appears to be irrelevant to the device characterization. It mainly links the device sensitivity to the number of pillars, not to the cause of high linearity nor any other claimed performance of the sensor itself.

Our response:

1. We have revised the title of Supplementary Note 2 to reflect the current focus of the theoretical analysis on enhanced sensitivity.
2. As discussed in the previous response letter, the deformation of the pyramid and changes in the contact area changes under pressure loadings are complex. Therefore, the theoretical analysis in the current study is only focused on mechanical simulation to help understand the improvement of the sensitivity due to gradient structures. We have also updated the manuscript to indicate that the theoretical analysis was used to help understand the sensor's enhanced sensitivity without bending (as opposed to explaining the complete set of performance parameters). Furthermore, the current theoretical result also indicates that a larger height difference and unit size can help increase the capacitance changes in the later compression stage (i.e., upon larger pressure loading) for maintaining the linear response over a large sensing range.

Although the theory presented in the current work gives a theoretical background to study the influence of gradient structures, it doesn't directly predict sensitivity and linear range, which will be our focus in future works. The combination of this theory with bending deformations is out of the scope of this paper and will be pursued in our future studies as well. We have added the limitation of the current methodology and future developments in the **Discussion**.

Our modification to the manuscript:

1. We have revised the title of Supplementary Note 2 to “**Note 2. Theoretical analysis of UPM and GPM at the dielectric/electrode interface to understand the enhanced sensitivity without bending**”.

2. The limitation of the current methodology has been added to the **Discussion**.

Further optimization of the laser parameters to create various height combinations in the microstructures and **the theoretical analysis to account for bending and to predict the other sensing parameters** will be pursued in our future works.

2. In A1, the influence of bending to the performance of the pyramids structure should be conducted through experiments or citing the theory from existing literatures.

Our response: We apologize for not including the experimental results in the previous response letter (which was included in the supplementary Fig. 9). The experimental results are now provided in this response letter.

Supplementary Fig. 9. **a** Optical image of the iontronic sensor bent over a diameter of 13 mm. **b** The pressure sensing performance and **c** schematic diagram of the iontronic pressure sensor under different bending conditions.

Our modification to the manuscript: None.

3. In A3, though the resolution of the sensor was calculated to be 0.0075%, the practical resolution seemed to be considerably poorer than that of the regular commercial weight scale. Thus, to compare the resolution/sensitivity of the proposed device with the commercial scale, the proposed measurement is inadequate. Even from a pressure sensor/load cell/scale with a reasonable sensitivity, the BCG signals can be easily recorded by using an LCR meter without additional filtering or signal processing capacity. Specifically, a number of prior arts have shown that the commercial scales, with a resolution of 0.1%, can record the typical BCG signals (e.g., Fig 6d) as well as respiratory patterns from human subjects (e.g., Nutrients 2022, 14, 2552).

<https://doi.org/10.3390/nu14122552>, IEEE Journal of Biomedical and Health Informatics, vol. 24, no. 1, pp. 69-78), which has not been observed in the manuscript nor in the attached video. Therefore, the claim that the resolution of the sensor outperforms that of the commercial weight scales may not be appropriate.

Our response: We thank the reviewer's additional suggestions.

1. We thank the reviewer for providing two references (10.3390/nu14122552 and 10.1109/JBHI.2019.2901635), which we carefully studied. However, we didn't find any evidence that "*the BCG signals can be easily recorded by using an LCR meter without additional filtering or signal processing capacity*" in both of these two references. In contrast, both references involved complex signal processing (either post-processing in the former or pre-processing in the latter). This is also supported by a review paper (Health Inf Sci Syst (2019) 7:10), which emphasizes the importance of the signal processing for BCG in the Conclusion "raw signal is noisy and nonstationary due to body movement, induced respiratory efforts, and the characteristics of the sensing system itself". Furthermore, there is no Figure 6d in the two suggested references. If the reviewer refers to another paper, please kindly let us know.

2. The resolution under ultrahigh pressure was calculated according to the method in the previously published papers (e.g., 0.02% (1 Pa under 5 kPa) in Nat Commun 12, 4731 (2021)) and compared with them. However, we acknowledge the challenge of directly comparing with the commercial weight scale, as it has gone through verification and validation processes for minimized device variation and improved stability. Because we do not have the needed resources to perform the BCG test, we have changed "outperforms" into "may outperform" when discussing the pressure resolution of the sensor in the **High pressure resolution under large pressure preloads**.

Our modification to the manuscript:

Note: All of the superscripts denote the reference in the manuscript.

4. The pressure resolution of 0.00725% under a preload of 2 MPa may outperform the commercial weight scale (Supplementary Fig. 10c), human skin (~ 7%)⁶⁸, and other microstructured pressure sensors previously reported in the literature^{9, 20, 43, 44}.

4. In A4, the method to loading and unloading using hand during the response time test could lead to highly unreliable or unrepeatable results. Thus, it is highly recommended that authors follow the standard practice in the published literatures, from which reliable data can be yielded (e.g., Nature Communications, 2014, 5: 3132).

Our response: We highly appreciate the reviewer's insightful comments. We have set up a linear actuator that applies and releases reliable and repeatable pressure (ca. 3~5 kPa with 1 second for application and 4 seconds for release) to measure the response time to a stable pressure, following the standard practice in the suggested and other more recent publications (Adv. Funct. Mater. 2019, 1808509; Science 2020, 370, 966-970; Nature communication 2022, 13:1317). We have uploaded Supplementary Movie 2 to show the similar response/recovery curve of the sensor in four consecutive cycles.

Our modification to the manuscript: We have uploaded Supplementary Movie 2 to show the similar response/recovery curve of the sensor in four consecutive cycles.

Besides the high sensitivity over the ultrabroad pressure range, the sensor with the gradient microstructures also exhibits a fast response/recovery of 6/11 ms from a pressure loading of 5 kPa (Fig. 5b, Supplementary Movie 2 and 3).